# Ecological Effects of *Sargassum fusiforme* Cultivation on Coastal Phytoplankton Community Structure and Water Quality: A Study Based on Microscopic Analysis

**DOI:** 10.3390/biology14070844

**Published:** 2025-07-10

**Authors:** Yurong Zhang, Rijin Jiang, Qingxi Han, Zimeng Li, Zhen Mao, Haifeng Jiao

**Affiliations:** 1College of Biological & Environmental Sciences, Zhejiang Wanli University, Ningbo 315100, China; zhangyurong@zwu.edu.cn (Y.Z.); 2024881020@zwu.edu.cn (Z.M.); 2Zhejiang Marine Fisheries Research Institute, Zhoushan 316021, China; 3School of Marine Sciences, Ningbo University, Ningbo 315211, China; hanqingxi@nbu.edu.cn; 4Marine Ecological and Environmental Monitoring Center of Zhejiang Province, Zhoushan 316021, China; 15257072451@163.com

**Keywords:** *Sargassum fusiforme* cultivation, microscopic quantitative analysis, water quality improvement, phytoplankton community dynamics

## Abstract

Seaweed farming is a green and sustainable way to produce food and to protect the marine environment. In this study, we examined how large-scale cultivation of *Sargassum fusiforme* affects seawater quality and tiny plant-like organisms in the ocean called phytoplankton. These organisms are important because they form the base of marine food chains. We collected water samples during both farming and non-farming seasons. The results showed that *S. fusiforme* farming helped raise oxygen levels and made nutrient concentrations, such as nitrogen and phosphorus, less excessive. Farming also helped reduce the overgrowth of certain dominant species and supported a more diverse and balanced phytoplankton community. We used microscopes to observe and to count the phytoplankton. Our findings show that seaweed farming can improve ocean health and can offer useful guidance for sustainable aquaculture and coastal ecosystem protection.

## 1. Introduction

As marine aquaculture continues to advance rapidly, ensuring its sustainability has become a critical issue. Sustainable aquaculture practices not only cater to the increasing need for seafood but promote the health of marine ecosystems [1,2]. Among various approaches, macroalgae cultivation has attracted much attention due to its ecological benefits. It helps improve water quality, reduce ocean acidification and eutrophication, enhance biodiversity, and boost marine carbon sequestration [3,4,5,6,7,8,9,10].

*Sargassum fusiforme* (Harvey) Setchell, 1931 is a species of macroalgae in the phylum Ochrophyta. It is mainly distributed in the warm-temperate coastal areas of the Western Pacific, especially in China, South Korea, and Japan [11]. Artificial cultivation of *S. fusiforme* began in Dongtou District, Zhejiang Province, in 1989. Since then, the farming area has rapidly expanded from 57.3 ha to over 867 ha, with a stable annual yield of 7000–9000 tons [12]. Dongtou District now accounts for more than 95% of the national total in terms of cultivation area, yield, and processed products, earning it the title “Home of *S. fusiforme* in China” [11,12].

*S. fusiforme* is not only an important edible marine vegetable but holds significant value in traditional Chinese medicine for its therapeutic properties [13]. It contains bioactive compounds, particularly fucoidans, which exhibit antitumor, antioxidant, antibacterial, and antiviral effects [14,15,16]. Studies have shown that *S. fusiforme* extracts can suppress harmful algal blooms through allelopathic effects [17,18]. The bioactive compounds released by the extracts can interfere with the photosynthesis and cellular functions of red tide microalgae [17,18]. Furthermore, α-linolenic acid, a compound from *S. fusiforme*, has been found to reduce chlorophyll fluorescence and electron transport in harmful dinoflagellates, suggesting that the growth of *S. fusiforme* could affect the structure of phytoplankton communities [19,20].

Phytoplankton play an essential role in marine primary production [21]. They are essential to the global carbon cycle and contribute to the biological carbon pump in the Earth’s oceans [22]. However, the excessive proliferation of phytoplankton can disrupt ecosystem balance and lead to the occurrence of harmful algal blooms (HABs). In particular, the abnormal proliferation of certain toxin-producing species, such as some dinoflagellates, can result in the mass mortality of marine organisms and pose significant health risks to humans throughout the food chain [18,23,24]. The composition and diversity of phytoplankton communities are strongly affected by environmental variations and shaped by a range of physical (e.g., temperature, light intensity, salinity), chemical (e.g., nutrient concentrations such as nitrogen and phosphorus), biological (e.g., grazing pressure, interspecific competition), and climatic factors (e.g., rainfall) [25,26,27]. As such, they are considered important biological indicators of changes in water environments, and can be used as ecological indices to assess the health of aquatic ecosystems [28,29,30].

Therefore, phytoplankton are not only key primary producers but important biological indicators for assessing changes in water environments and ecosystem health [11,17]. With the rapid expansion of macroalgae farming, there is a growing need to manage its ecological impacts more effectively. Ecological management in seaweed cultivation involves not only maintaining stable yields, but ensuring that farming practices do not disrupt local biodiversity, nutrient balance, or water quality [18]. In particular, understanding how *S. fusiforme* cultivation influences phytoplankton communities can help identify early signs of ecological imbalance and guide sustainable farming decisions.

High-throughput sequencing has been widely used to study phytoplankton communities, offering rich genetic and structural information. However, it lacks quantitative accuracy and cannot fully reflect actual phytoplankton abundance [29,31,32]. In our 2022 study, we used high-throughput sequencing combined with a redundancy analysis (RDA) and phylogenetic molecular ecological networks (pMENs) to examine the effects of *S. fusiforme* cultivation on community structure [33]. To improve the accuracy of quantitative analysis, we used microscopy-based enumeration to investigate phytoplankton communities and water quality in different areas and at different time periods. Based on the data collected in April and June of 2018 and 2019, we assessed the ecological effects of large-scale *S. fusiforme* cultivation on coastal phytoplankton dynamics and the related environmental factors. The findings offer insights into the ecological role of *S. fusiforme* and support its sustainable use in marine ecosystem management.

## 2. Materials and Methods

### 2.1. Sample Collection

#### 2.1.1. Sampling Site Selection

The sampling was at the Banpingshan cultivation base, located in the southeastern part of Dongtou Island, Wenzhou City, Zhejiang Province (E27°41′19″–28°01′10″, N120°59′45″–121°15′58″). The average water depth at the sampling site was 9.5 ± 0.36 m. *S. fusiforme* is cultivated using longline methods. Seedlings are introduced each July and harvested the following April. The cultivation base covers an area of about 300 ha, with an annual yield of about 2000 tons, establishing it as one of the primary *S. fusiforme* cultivation hubs in Dongtou District.

Samples were collected in April and June of 2018 and 2019 to assess how *S. fusiforme* cultivation affects phytoplankton communities. April represents the harvest season for *S. fusiforme*, when the biomass peaks [18,33]. At this time, the influence of cultivation on the water environment is most significant. June, approximately two months after the harvest, represents a transitional period between two cultivation cycles, as new *S. fusiforme* seedlings are typically transplanted in July. During this interval, the previously cultivated seaweed has been completely removed, and no new biomass has been introduced. Thus, June served as a post-harvest control point. This allowed us to assess the environmental conditions and phytoplankton community structure without the seaweed. It helped distinguish seasonal variation from cultivation effects. Samples were collected at both the harvest peak in April and the post-harvest period in June. This approach also allowed us to assess how seasonal climate and hydrological changes, such as rainfall, runoff, and salinity fluctuations, among others, may influence phytoplankton community structure.

A total of 12 sampling stations were set up, and 288 water samples were collected. Each station was located within or near the *S. fusiforme* farming area. Six parallel samples were collected at each station. To assess the impact of farming activities, the stations were divided into the following three zones based on their distance from the cultivation area (Figure 1):

Cultivation area A (stations A1–A6): Located inside the cultivation area. This zone was used to evaluate the direct effects of cultivation on water quality.

Adjacent area B (stations B1–B3): Located 200–500 m from the cultivation area. No cultivation was conducted here, but the area may have been indirectly affected.

Control area C (stations C1–C3): Located about 2000 m offshore. This zone was considered the control area, representing natural background conditions without cultivation disturbance.

This spatial division was based on the method described by Chai et al. (2018) [34]. At each station, water samples were taken from two depths: surface (0.5 m below surface) and bottom (0.5 m above seafloor).

#### 2.1.2. Sample Collection and Determination of Physical and Chemical Parameters

Water samples (2 L each) were collected from both the surface and bottom layers at each station using a Plexiglass water sampler (Fushun Bright Science and Technology Co., Ltd., Fushun, China). One liter from each sample was transferred into a 2.5 L gallon bucket. After collection, one 1 L sample was fixed with Lugol’s iodine for later analysis of phytoplankton community structure. The other 1 L sample was used to measure nitrite–nitrogen (NO_2_–N), nitrate–nitrogen (NO_3_–N), ammonium–nitrogen (NH_4_–N), total nitrogen (TN), phosphate–phosphorus (PO_4_–P), total phosphorus (TP), and silicate–silicon (SiO_3_–Si) using a SEAL Analytical AA3 flow injection analyzer (Norderstedt, Germany). The analyses were conducted in accordance with the Chinese national environmental standard HJ 442.3-2008 [35]. The method detection limits (MDLs) for each parameter were as follows: 0.001 mg/L for NO_2_–N, NO_3_–N, NH_4_–N, and PO_4_–P; 0.002 mg/L for SiO_3_–Si; 0.05 mg/L for TN; and 0.03 mg/L for TP.

Additionally, seawater temperature (T), dissolved oxygen (DO), pH, and salinity (Sal) were recorded on site using multi-function water quality meters (YSI Inc., Yellow Springs, OH, USA): T and Sal were measured with the EC300 A (Jiaxing, China), pH with the pH 100 A, and DO with the 550 A. The water depth was measured using a COMPACT-CTD (NBOSI Ocean Sensors, Norderstedt, Germany).

### 2.2. Phytoplankton Sample Analysis

This study employed microscopic examination techniques for the identification and quantitative analysis of phytoplankton. The 1 L water samples collected were concentrated to 10 mL for subsequent microscopic observation. For each sample, a volume of 2 mL was used and analyzed in triplicate. Phytoplankton abundance was expressed as the number of cells per liter (cells/L). Species identification and cell counting were performed using an inverted microscope (Leica DMi8 (Leica Microsystems, Wetzlar, Germany)). All procedures followed the “Specifications for Oceanographic Survey” (GB/T 12763.6-2007) [36]. Species identification was based on morphological characteristics with reference to the *Flora Algarum Marinarum Sinicarum*, Vol 5: *Bacillariophyta* [37] and Vol 2: *Dinophyta* [38]. Species names were verified and updated using the latest nomenclature from AlgaeBase (Galway, UK) and the World Register of Marine Species (WoRMS) (Oostende, Belgium).

### 2.3. Data Processing and Analysis

The spatial distribution of the sampling stations was visualized using ArcGIS 10.3. Phytoplankton abundance percentages were illustrated in a bar chart created with Origin 9.2. The biodiversity indices, including the Shannon–Wiener diversity (*H*′), Margalef’s richness (*D*), and Pielou’s evenness index (*J*), were computed using Primer 6.0. Additionally, Primer 6.0 was utilized to apply square root transformation to the phytoplankton abundance data and to calculate the Bray–Curtis dissimilarity index. One-way ANOSIM was employed to assess the community structure of the phytoplankton.

The species with dominance values (*Y*) ≥ 0.01 were defined as dominant and included in the analysis.

*Y* was calculated as follows [39]:(1)$$Y=\left(\frac{ni}{N}\right)\times fi,$$
where n_i_ is the abundance of individuals of species, N is the total abundance of individuals of all species in all samples, and f_i_ is the occurrence frequency of the species i among the samples.

One-way ANOVA was conducted using SPSS 20.0 to evaluate the differences in the water quality parameters, phytoplankton abundance, diversity indices (*H*′, *D*, and *J*), and environmental factors across the various sampling locations. Prior to ANOVA, the Shapiro–Wilk test was used to assess data normality, and Levene’s test was performed to evaluate the homogeneity of variances. Only when both assumptions were satisfied (*p* > 0.05), Tukey’s HSD post hoc test was conducted.

A significance threshold of *p* < 0.05 was used.

To further explore the relationships between phytoplankton community structure and environmental factors, this study introduced multivariate ecological statistical methods. These methods enhanced the explanatory power of the analysis and improved the credibility of the results.

First, we used Canoco 5.0 to model the relationships between the dominant phytoplankton species and environmental variables. To determine the type of community response to environmental factors, we conducted a detrended correspondence analysis (DCA) on the data from all four seasons. The gradient lengths were all less than 3. This indicated that a linear ordination model was suitable. Therefore, we selected a redundancy analysis (RDA) to visualize the relationships between the phytoplankton community and environmental variables.

Next, to verify the statistical significance of the RDA model and the environmental variables, we performed a Monte Carlo permutation test (*n* = 999). This non-parametric test is well suited for small-sample ecological data. It allowed us to assess whether the model was significantly better than a random structure and to enhance the robustness of the results.

In addition, to further identify the co-occurrence patterns and ecological response mechanisms between dominant phytoplankton groups and key environmental variables, we applied a phylogenetic molecular ecological network analysis (pMEN). This method constructs networks based on correlations among species. It reveals potential cooperative relationships and extracts topological properties (such as connectivity and modularity). In this way, it identifies key environmental drivers and core responsive groups.

The environmental factors included NH_4_–N, NO_2_–N, NO_3_–N, PO_4_–P, TN, TP, T, DO, pH, and Sal. Network construction and analysis were conducted using the MENA platform (http://ieg2.ou.edu/MENA/, accessed on 28 January 2022).

### 2.4. Reference Information on High-Throughput Sequencing

The high-throughput sequencing data used for comparison were taken from our earlier study (Zhang et al., 2022 [33]). In that study, we used the Illumina MiSeq platform (PE250/PE300) with 18S rRNA V4 region primers (18sV4F and 18sV4R). Full details of the sequencing method, primers, and data processing steps are provided in the published article. In this manuscript, those results are only used for comparison with microscopy-based analysis.

## 3. Results

### 3.1. Water Quality

#### 3.1.1. April 2018 (Cultivation)

In April 2018, the pH in the cultivation area (8.38 ± 0.02) was significantly higher than in the control area (8.21 ± 0.03, *p* < 0.05), but not significantly different from the adjacent area (8.27 ± 0.05, *p* > 0.05). Likewise, the DO levels in the cultivation area (306.3 ± 3.1 μmol/L) were higher than in the control area (296.9 ± 6.3 μmol/L, *p* < 0.05) and similar to the adjacent area (300.0 ± 3.1 μmol/L, *p* > 0.05). The concentrations of NO_3_–N (5.7 ± 0.9 µmol/L), PO_4_–P (0.2 ± 0.0 µmol/L), TP (0.7 ± 0.2 µmol/L), and SiO_3_–Si (5.5 ± 1.0 µmol/L) were all significantly lower in the cultivation area than in the adjacent area (NO_3_–N: 9.4 ± 1.4 µmol/L; PO_4_–P: 0.3 ± 0.1 µmol/L; TP: 1.1 ± 0.2 µmol/L; SiO_3_–Si: 7.3 ± 0.3 µmol/L, *p* < 0.05), while no significant differences were found between the cultivation area and the control area (*p* > 0.05). The NH_4_–N (9.3 ± 1.8 µmol/L) and TN (224.8 ± 32.1 µmol/L) in the cultivation area were slightly higher than those in the adjacent area (NH_4_–N: 7.1 ± 1.3 µmol/L; TN: 203.4 ± 35.7 µmol/L) and the control area (NH_4_–N: 8.4 ± 1.8 µmol/L; TN: 221.3 ± 552.1 µmol/L), but these differences were not significant (*p* > 0.05) (Figure 2). In the cultivation area, concentrations of NO–N, PO–P, TP, and SiO_3_;–Si were reduced by 39.4%, 39.3%, 36.4%, and 24.7%, respectively, compared to the adjacent area. Relative to the control area, SiO–Si decreased by 27.6%, while pH and DO increased by 2.1% and 3.2% (Figure 2). These results indicate that *S. fusiforme* cultivation increased pH and DO and reduced nutrient concentrations, suggesting a significant improvement in water quality during the cultivation season.

#### 3.1.2. June 2018 (Non-Cultivation)

In June 2018, the DO and pH in the cultivation area (256.3 ± 3.0 µmol/L, 8.05–8.08) showed no significant differences from those in the adjacent area (256.3 ± 3.0 µmol/L, approximately 8.06) and the control area (256.3 ± 3.0 µmol/L, approximately 8.07, *p* > 0.05). The NO_3_–N in the cultivation area (23.7 ± 10.4 µmol/L) was slightly higher than in the adjacent (19.6 ± 0.4 µmol/L) and control areas (20.4 ± 0.9 µmol/L), but these differences were not significant (*p* > 0.05). The NO_2_–N (2.3 ± 0.2 µmol/L) showed no significant difference from the adjacent area (2.1 ± 0.2 µmol/L, *p* > 0.05), but was significantly higher than the control area (1.9 ± 0.1 µmol/L, *p* < 0.05). The NH_4_–N (6.9 ± 2.5 µmol/L) and the PO_4_–P (1.0 ± 0.1 µmol/L) were significantly higher in the cultivation area than in the adjacent (NH_4_–N: 4.1 ± 0.3 µmol/L; PO_4_–P: 0.7 ± 0.3 µmol/L) and control areas (NH_4_–N: 4.0 ± 0.2 µmol/L; PO_4_–P: 0.8 ± 0.1 µmol/L, *p* < 0.05), and no significant differences were observed between the adjacent and control areas (*p* > 0.05; Figure 2). Compared with the adjacent area, NO_3_–N, NH_4_–N, and PO_4_–P in the cultivation area increased by approximately 20.9%, 68.3%, and 42.9%, respectively (Figure 2). These results suggest that, after harvest, the removal of nutrients by *S. fusiforme* weakened, allowing NO_3_–N, NO_2_–N, NH_4_–N, and PO_4_–P to accumulate, thereby reducing the water purification capacity during the non-cultivation period.

#### 3.1.3. April 2019 (Cultivation)

In April 2019, the pH (8.21 ± 0.00) and DO (325.0 ± 9.4 µmol/L) in the cultivation area were significantly higher than those in the adjacent area (pH: 8.10 ± 0.02; DO: 315.6 ± 3.1 µmol/L) and the control area (pH: 8.13 ± 0.03; DO: 315.6 ± 3.1 µmol/L, *p* < 0.05). The NO_3_–N (33.8 ± 0.9 µmol/L) and the NO_2_–N (0.6 ± 0.6 µmol/L) were significantly lower in the cultivation area than in the control area (NO_3_–N: 36.2 ± 2.3 µmol/L; NO_2_–N: 1.9 ± 1.2 µmol/L, *p* < 0.05), and showed no significant differences from the adjacent area (NO_3_–N: 33.3 ± 0.8 µmol/L; NO_2_–N: 0.6 ± 0.7 µmol/L, *p* > 0.05). The TN (220.6 ± 22.8 µmol/L) was also significantly lower in the cultivation area than in the control area (237.7 ± 25.7 µmol/L, *p* < 0.05) and similar to the adjacent area (219.1 ± 18.6 µmol/L, *p* > 0.05), while the PO_4_–P showed no significant differences among all three areas (*p* > 0.05; Figure 3). Compared with the control area, the NO_3_–N, NO_2_–N, and TN in the cultivation area decreased by approximately 6.7%, 68.4%, and 6.5% (Table 1), respectively, indicating that *S. fusiforme* cultivation enhanced the DO and pH and reduced the NO_3_–N, NO_2_–N, and TN concentrations (Figure 3), demonstrating a clear water quality improvement during the cultivation season.

#### 3.1.4. June 2019 (Non-Cultivation)

In June 2019, the pH (8.20–8.27) and DO (290.6 µmol/L) showed no significant differences among the cultivation, adjacent, and control areas (*p* > 0.05). The NO_3_;–N in the cultivation area (26.8 ± 14.8 µmol/L) was not significantly different from that in the adjacent (20.5 ± 0.6 µmol/L) and control areas (20.7 ± 1.1 µmol/L, *p* > 0.05). The NO_2_–N (2.4 ± 0.3 µmol/L) and the NH_4_–N (8.6 ± 2.4 µmol/L) were significantly higher in the cultivation area than in the adjacent (NO_2_–N: 0.7 ± 0.3 µmol/L; NH_4_–N: 6.8 ± 0.2 µmol/L) and control areas (NO_2_–N: 0.6 ± 0.2 µmol/L; NH_4_–N: 6.6 ± 0.2 µmol/L, *p* < 0.05), while the PO_4_–P (1.0 ± 0.1 µmol/L) was significantly lower in the cultivation area than in the adjacent (1.4 ± 0.3 µmol/L) and control areas (1.5 ± 0.1 µmol/L, *p* < 0.05), and there was no significant difference between the adjacent and control areas (*p* > 0.05; Figure 3). Compared with the adjacent area, the NO_3_–N and NH_4_–N in the cultivation area increased by approximately 30.7% and 26.5%, and by 30.0% and 30.3% relative to the control area (Figure 3). These results suggest that, after harvest, the nutrient removal weakened and NO_3_–N and NH_4_–N accumulated again, reducing the water purification capacity during the non-cultivation period.

Overall, the data from April (cultivation period) and June (non-cultivation period) in 2018 and 2019 show that *S. fusiforme* cultivation increased the pH and DO and decreased the NO_3_–N, NO_2_–N, PO_4_–P, TP, and SiO_3_–Si, indicating that *S. fusiforme* cultivation improved coastal water quality. By contrast, during the non-cultivation period, nutrient concentrations increased due to the absence of algal biomass, suggesting that the water purification capacity declined without cultivation. These findings demonstrate the significant ecological restoration potential of *S. fusiforme* cultivation in coastal waters.

In both years, the salinity increased from April to June across all zones (from 27.9–27.7‰ in April to 28.4–28.6‰ in June 2018, and from 26.5‰ in April to 28.3–28.4‰ in June 2019) (Figure 2 and Figure 3), consistent with seasonal changes, such as reduced rainfall and enhanced evaporation.

### 3.2. Phytoplankton Species

In 2018 and 2019, 79 species of phytoplankton belonging to five phyla and 47 genera were observed. This included 27 genera and 54 species of Bacillariophyta, 13 genera and 18 species of Dinophyta, 2 genera and 2 species of Chrysophyta, 2 genera and 2 species of Cyanophyta, and 3 genera and 3 species of Chlorophyta. Diatoms and dinoflagellates were the dominant phytoplankton groups across all regions and sampling times (Figure 4).

### 3.3. Phytoplankton Abundance

No significant difference in phytoplankton abundance was observed among the different areas sampled in April (*p* > 0.05). In June, the abundance in the cultivation area (15,982 ± 9531 cells/L) was significantly lower than in the adjacent area (88,373 ± 10,855 cells/L) and the control area (358,673 ± 136,877 cells/L) (*p* < 0.05; Table 1). Overall, phytoplankton abundance increased markedly in June compared to April, particularly in the control and adjacent areas.

### 3.4. Dominant Phytoplankton Species

*Skeletonema costatum* was consistently the dominant species in all four surveys, across the cultivation, adjacent, and control areas, although its dominance varied by region and season. In April 2018, its dominance was significantly lower in the cultivation area compared to the control (*p <* 0.05; Table 2). By June 2018, no significant differences were detected between the cultivation and adjacent areas (*p* > 0.05; Table 2). In June 2018, Dinophyta abundance and species increased notably at area A and B. *Scrippsiella acuminata* was the most dominant species, with dominance values of 0.47 at area A and 0.42 at area B. In April 2019, *S. costatum* exhibited significantly lower dominance in the cultivation area compared to the control (*p* < 0.05; Table 3). However, in June 2019, no significant differences was observed among the areas (*p* > 0.05). Overall, in April, the cumulative dominance of phytoplankton was lower in the cultivation area than in the adjacent and control areas, while no significant differences were observed in June (*p* > 0.05; Table 3).

### 3.5. Phytoplankton Diversity

At the same sampling time points, the diversity indices (*H*′, *D*, and *J*) showed little difference between the cultivation and adjacent areas. However, both areas exhibited significantly higher diversity values than the control (*p* < 0.05) (Table 4). Within the same year, the diversity indices at each site were significantly higher in April than in June (*p* < 0.05; Table 4). In both 2018 and 2019, diversity indices were significantly higher in April than in June (*p* < 0.05; Table 4).

### 3.6. Similarity Analysis of Community Composition

One-way ANOSIM showed that, in April 2018, the phytoplankton community structure differed significantly between the cultivation area and the control area (R = 0.667, *p* = 0.024), while no significant differences were observed among the three areas in June (all *p* > 0.05; Table 5). A similar seasonal pattern was found in 2019 (R = 0.537, *p* = 0.048), consistent with the results of 2018 (Table 5).

### 3.7. Relationship Between Dominant Phytoplankton Species and Environmental Factors

The RDA identified N, P, Si, and DO as the critical factors influencing dominant phytoplankton species. In April 2018, *S. costatum* density was positively correlated with N, P, and Si, but negatively with DO. In June 2018, the densities of *S. costatum*, *T. muelleri*, *Protoperidinium pentagonum*, *Tripos furca*, and *Dictyocha fibula* were positively associated with DO and SiO_3_–Si, but negatively with N and P (Figure 5). In April 2019, *S. costatum* density was positively correlated with NO_3_–N and SiO_3_–Si, but negatively with NH_4_–N, NO_2_–N, TN, PO_4_–P, and DO. Similarly, in June 2019, *S. costatum* density was negatively correlated with TN, P, Si, and DO (Figure 6).

Although the RDA ordination plots revealed certain associations between dominant phytoplankton groups and environmental variables, the results of the Monte Carlo permutation tests showed that the overall RDA models for all four sampling periods were not statistically significant (*p* > 0.05). This may be partly attributed to the relatively limited sample size (*n* = 6 per group), which could reduce the statistical power to detect significant multivariate relationships. This indicates that, under the current sampling design and spatial resolution, the combined effects of the measured environmental variables on phytoplankton community structure were not significantly stronger than those expected by chance. However, the results of single-variable analyses revealed that several key nutrient factors still exhibited strong explanatory power for community variation during specific seasons.

In April 2018, SiO_3_;–Si (R^2^ = 0.285, *p* = 0.001) and TN (R^2^ = 0.268, *p* = 0.002) were significantly correlated with phytoplankton community structure, suggesting that, under nutrient-rich spring conditions, these factors likely played a dominant role in determining species composition and abundance patterns. In June 2018, NO_3_–N accounted for an explanatory power of R^2^ = 0.194 with a *p*-value of 0.065, which, although not statistically significant, was near the threshold, indicating a potential influence on community structure during the late spring to early summer transition. Similarly, in June 2019, silicate SiO_3_–Si (R^2^ = 0.165, *p* = 0.056) and NO_2_;–N (R^2^ = 0.163, *p* = 0.072) also approached significance, suggesting that under strongly stratified summer conditions with limited nutrient regeneration, these factors may have played a substantive role in regulating shifts in community composition (Table 6).

The relationships between dominant phytoplankton species and environmental variables (NH_4_–N, NO_2_–N, NO_3_–N, PO_4_–P, SiO_3_–Si, TN, TP, DO, pH, Sal, and T) were analyzed using a pMEN analysis based on Mantel’s tests. The significance of correlations was determined from unadjusted *p*-values (*p* < 0.05). Additionally, the pMENs were constructed for dominant species across the four surveys (Figure 6 and Figure 7). *Planktoniella blanda* was significantly correlated with TN, pH, PO_4_–P (*p* < 0.01), and TP (*p* < 0.05) in April 2018 (Figure 7a). In June 2018, *S. costatum* was significantly correlated with PO_4_–P (*p* < 0.05). *A. tamarense* showed significant correlations with TN and PO_4_–P (*p* < 0.05). Both *D. fibula* and *Tripos muelleri* were significantly associated with NO_3_;–N (*p* < 0.05) (Figure 7b). In April 2019, *S. costatum* was significantly correlated with pH (*p* < 0.01) and SiO_3_–Si (*p* < 0.05) (Figure 8a). In June 2019, it remained significantly correlated with pH (*p* < 0.05). *Paralia sulcata* showed significant correlations with TN and TP (*p* < 0.05) (Figure 8b).

## 4. Discussion

### 4.1. S. fusiforme Cultivation Improves Water Quality

This study used microscopic observation to analyze phytoplankton samples collected in the spring and summer of 2018 and 2019. The goal was to assess the impact of *S. fusiforme* cultivation on phytoplankton diversity and water quality. A comparison of water quality in April (cultivation period) and June (non-cultivation period) of 2018 and 2019 showed that *S. fusiforme* cultivation effectively increased pH and DO. It also reduced the concentrations of NO_3_–N, NO_2_–N, PO_4_–P, TP, and SiO_3_–Si. These changes led to better water quality. However, after harvest, the absence of *S. fusiforme*-driven nutrient removal weakened this nutrient mitigation. As a result, nutrient concentrations accumulated again during the non-cultivation period.

These findings are consistent with our 2022 study that used high-throughput sequencing, although that study only included data from 2019. By adding the 2018 data, this study further confirms the effectiveness of *S. fusiforme* cultivation in improving water quality [33].

Although some nutrient concentrations observed in this study, especially nitrogen and silicate, were relatively high, they still fell within the background range for the region. According to Ye et al. (2020), the typical spring and summer concentrations of DIN, DIP, and DSi in the nearshore East China Sea range from 5.3 to 11.9, 0.31 to 0.53, and 11.1 to 23.2 µmol/L, respectively [40]. The values measured in this study were within these ranges, and reflect the naturally high nutrient levels in the nearshore East China Sea. In addition, the phytoplankton biomass did not always increase when the nutrient levels were high. This suggests that algal blooms are not only caused by nutrient enrichment. Other environmental factors, such as water temperature, also play an important role. As shown in Figure 2 and Figure 3, the water temperature increased to about 25 °C in June 2018, which coincided with the highest phytoplankton abundance recorded in this study. During that month, the abundance in the adjacent area (88,373 ± 10,855 cells/L) and the control area (358,673 ± 136,877 cells/L) rose sharply (Table 1), approaching the commonly accepted bloom threshold of 10^5^ cells/L [41]. By contrast, in the other three surveys, the water temperatures were noticeably lower, and no significant increase in phytoplankton was observed, despite high nutrient levels. These results suggest that nutrient enrichment alone is not sufficient to trigger algal blooms. Suitable water temperatures are also necessary.

Although *S. fusiforme* can absorb and remove nutrients, local hydrodynamics and other environmental factors may cause temporary nutrient accumulation or fluctuations. For example, in April 2018 during the cultivation period, NH_4_–N and TN concentrations were slightly higher in the cultivation area than in the adjacent and control areas. In June 2019, after harvest, PO_4_–P was significantly lower in the cultivation area than in the adjacent and control areas. Factors such as rainfall runoff, sediment resuspension, and nearshore discharges can also introduce nutrients into the water column. These processes may contribute to variations in nutrient concentrations across space and time [42]. Thus, these abnormal nutrient patterns may reflect a combined influence of biological removal and local environmental processes.

Although *S. fusiforme* itself does not directly consume silicate, SiO_3_–Si concentrations in the cultivation area remained lower than those in the adjacent and control areas. The reasons for this pattern are not yet clear. Factors such as local hydrodynamics, plankton community composition, or other unmeasured ecological processes may help explain this result and require further study.

In addition, salinity was slightly lower at all stations in April and increased markedly by June. This pattern was closely related to the seasonal changes in freshwater runoff and increased evaporation [43]. These environmental factors also provide a background context for the re-accumulation of nutrients during the non-cultivation period. These results highlight the potential of *S. fusiforme* cultivation as an effective nature-based solution for improving water quality in coastal ecosystems.

### 4.2. Impact of S. fusiforme Cultivation on Phytoplankton Community

Microscopic observation is well-suited for the quantitative analysis of larger phytoplankton, enabling precise assessment of the abundance and distribution of larger diatom and dinoflagellate groups (Table 7). Although *S. fusiforme* cultivation did not markedly change the overall phytoplankton composition, it significantly affected the distribution of the dominant species. In the April survey, although the phytoplankton abundance did not differ significantly among the sea areas, large-scale *S. fusiforme* cultivation significantly reduced the proportion of the dominant species *S. costatum.* This suggests that *S. fusiforme* may affect phytoplankton growth through nutrient competition, shading, or allelopathy [19,44,45,46]; however, this phenomenon was not found by our previous research using high-throughput sequencing analysis [33].

It is worth noting that changes in the dominant species are influenced not only by cultivation activities but by temporal environmental factors. For example, in June 2018, a significant increase in Dinophyta abundance was observed at area A and B, with *S. acuminata* identified as the primary dominant species. This pattern may be related to temporal warming, which enhances water column stratification and reduces vertical mixing, creating favorable conditions for dinoflagellates with vertical migration ability [47,48,49,50,51]. In addition, *S. acuminata* can rapidly develop from resting cysts under suitable temperatures [51,52]. The warm, nutrient-rich waters near areas A and B likely triggered cyst germination and rapid population growth. By contrast, area C is located in a control area with strong hydrodynamic conditions and fewer cyst deposits, making dinoflagellate blooms less likely. These results highlight the key role of temperature and water structure in shaping the dominant species, and demonstrate the importance of microscopy in detecting species-level changes. These interannual differences may be partly explained by large-scale climate anomalies.

These interannual differences may be partly explained by large-scale climate anomalies. According to the China National Climate Center, 2019 was an official El Niño year. From June 2018, sea surface temperatures in the central and eastern Pacific gradually rose, and El Niño conditions developed by September 2018, continuing into mid-2019. During this period, southern China experienced extended overcast and rainy weather, with reduced sunlight, frequent rainfall, and occasional low temperatures [53]. These conditions likely affected seawater temperature, light availability, and nutrient levels in the study area, which may have influenced the phytoplankton growth and community structure. In addition, as shown in Figure 2, the seawater temperature in April 2018 was about 3 °C higher than that in April 2019, which may have further contributed to the observed differences between the two years. Such environmental changes could have weakened the effects of *S. fusiforme* cultivation or altered species responses, helping to explain the differences observed between 2018 and 2019.

In addition to climatic influences, regional ecological conditions may also play an important role in shaping phytoplankton community structure. The coastal waters examined in our study are located near the Nanji Islands. A year-round investigation by Li et al. (2010) recorded only 80 phytoplankton species, primarily diatoms and dinoflagellates, which aligns well with our findings [54]. Their study also reported an average cell abundance of 1.03 × 10^6^ cells/L, with notable temporal variation—higher in the spring and summer, and lower in the autumn and winter. The Shannon–Wiener index (H′) ranged from 0.32 in winter to 1.75 in autumn, indicating that moderate diversity and low abundance are common in this region.

Similarly, the 2017 Ecological Bulletin for the Coastal Waters of Wenzhou reported 129 species, suggesting consistent diversity over time [55]. By contrast, Guo et al. (2014) and Shen et al. (2023) identified 242 and 118 species, respectively, across larger and more diverse marine environments [56,57]. Considering the spatial and temporal scope of our study, the observed species richness and cell abundance are ecologically reasonable and consistent with the previous regional findings.

During the cultivation period in April, there were no significant differences in the total phytoplankton abundance among the three areas. However, the biodiversity indices (*H′*, *D*, and *J*) were significantly higher in the cultivation and adjacent areas than in the control area. The dominance of *S. costatum* was also significantly lower in the cultivation area. Specifically, its dominance decreased from 0.74 in the control area to 0.17 in the cultivation area in April 2018, and from 0.19 to 0.06 in April 2019. These results suggest that *S. fusiforme* cultivation may reduce the dominance of a single species and improve the evenness and diversity of the phytoplankton community. This finding is consistent with our previous conclusions based on high-throughput sequencing, comparing phytoplankton α and β diversity (Table 7) [33]. Thus, in the cultivation area, the increased phytoplankton diversity enriches the food web and enhances ecosystem stability. Relatively rich species diversity generally supports greater resilience and disturbance resistance, as species interactions in diverse community help distribute ecological stress, reducing the risks of single-species overgrowth or collapse [58,59,60,61]. Our findings further support this notion, as the elevated phytoplankton diversity observed in the cultivation area was associated with a more complex and interdependent food web, promoting overall ecological balance and stability [62,63].

Moreover, higher phytoplankton diversity is often linked to a lower risk of harmful algal blooms (HABs). In diverse communities, species compete more intensely for resources, which can suppress the rapid growth of harmful species [60,61,64]. This is consistent with local observations from seaweed farmers, who reported fewer red tide events in areas where *S. fusiforme* was cultivated. A microscopic examination also revealed several dinoflagellate species with known toxic potential. These included *Alexandrium* sp. and *Dinophysis caudata*, which are known to produce paralytic shellfish poisoning (PSP) and diarrhetic shellfish poisoning (DSP) toxins, respectively [65,66]. However, these species were found in low abundance, and no HAB events occurred during the study period. In addition, no cyanobacteria or toxic diatoms were detected. These results suggest that *S. fusiforme* cultivation did not promote the growth of toxin-producing phytoplankton, and may help reduce HAB risks by maintaining a diverse and stable phytoplankton community.

One-way ANOSIM further confirmed significant differences in the phytoplankton community structure between the cultivation and non-cultivation areas. These structural differences reflect the environmental impact of cultivation. They also suggest that cultivation activities may help control harmful algal blooms (HABs), since increased diversity and stability are often linked to reductions in blooms such as red tides [48,49]. In this study, the RDA, Monte Carlo permutation tests, and pMEN analysis similarly revealed the key environmental factors affecting the dominant species of phytoplankton, including N, P, Si, and DO. These findings align with our previous high-throughput sequencing results, supporting the significant impact of environmental factors on phytoplankton community structure [33]. Among these factors, SiO_3_–Si is particularly important as it serves as a limiting nutrient for the growth of siliceous phytoplankton, especially diatoms. Its availability can strongly influence community composition and succession patterns in coastal ecosystems [67]. In this study, the RDA results from April 2018 showed that SiO_3_–Si was significantly correlated with phytoplankton community structure (R^2^ = 0.285, *p* = 0.001), highlighting its ecological role in shaping phytoplankton communities. Therefore, variations in the SiO_3_–Si levels, even though *S. fusiforme* does not directly utilize silicate, may reflect shifts in the phytoplankton composition and nutrient cycling processes within the system. Taken together, our quantitative microscopic analysis demonstrates that large-scale *S. fusiforme* cultivation significantly influences the structure and diversity of the phytoplankton communities. This finding is consistent with our previous results.

### 4.3. Comparison of Microscopic Observation and High-Throughput Sequencing in Analyzing Phytoplankton Communities

Although this study employs analytical tools similar to those used in our previous research [33], such as RDA and pMEN, its innovation lies in the integration of microscopy data with an expanded temporal dimension. The analyses of the diversity indices, RDA, and pMEN based on both microscopic observation and high-throughput sequencing indicate that large-scale *S. fusiforme* cultivation enhances phytoplankton community diversity. Specifically, these analytical methods collectively reveal the positive impact of *S. fusiforme* cultivation on the structure of phytoplankton communities, not only increasing community diversity indices but optimizing the distribution patterns of phytoplankton within the ecosystem (Table 7). Therefore, this study verifies, from multiple analytical perspectives, that *S. fusiforme* cultivation plays an important role in enhancing phytoplankton diversity and promotes the stability of the aquatic environment.

Microscopic observation is particularly well suited for quantifying larger phytoplankton. It allows for an accurate assessment of the abundance and distribution of large diatoms and dinoflagellates, such as *P. blanda*, *Coscinodiscopsis jonesiana*, *P. sulcata*, and *Pleurosigma pelagicum*. These data provide reliable information on the spatial distribution of the dominant species in this study. By contrast, high-throughput sequencing revealed more small and difficult-to-identify groups, such as *Micromonas pusilla* (Chlorophyta, approximately 0.2–0.3 µm in size) [68], *Bathycoccus prasinos* (Chlorophyta, approximately 1–3 µm) [69], and *Aureococcus anophagefferens* (Ochrophyta, approximately 1.5–2 µm) [33,70]. These sequencing methods have enriched the genetic information on phytoplankton communities, revealing the diversity of small phytoplankton in the water, and are especially useful for detecting small communities in complex aquatic systems [71,72]. Therefore, microscopic observation provides important information on the abundance and spatial distribution of the dominant species, while high-throughput sequencing supplements data on the diversity of small species (Table 7) [73,74]. However, high-throughput sequencing of 18S rRNA amplicons did not adequately identify the phytoplankton at the species level [33]. This discrepancy may also partially result from the choice of the sequencing platform and primer sets. For example, the Illumina MiSeq PE250/PE300 platform and the 18S V4 universal primers (18sV4F: 5′-CCAGCASCYGCGGTAATTCC-3′ and 18sV4R: 5′-ACTTTCGTTCTTGATYRA-3′) may introduce amplification biases. These biases can affect the estimated relative abundances of certain taxonomic groups. The combination of microscopy and high-throughput sequencing provides a more comprehensive understanding of the phytoplankton community structure. It also offers deeper ecological insights into how environmental factors influence community dynamics.

When interpreting changes in the abundance of certain dominant species, differences may exist between the results of microscopic observation and high-throughput sequencing. For example, in the case of *S. costatum*, the microscopic data showed a significant decrease in its abundance in the cultivation area, while high-throughput sequencing did not reveal the same pattern. This discrepancy may result from the differences in the detection sensitivity, sample processing, and community structure analysis between the two methods.

First, detection sensitivity is closely related to cell size. High-throughput sequencing can efficiently detect picoplankton and nanoplankton. These small phytoplankton often cannot be reliably identified under traditional light microscopy due to their tiny size and lack of distinct morphological features. By contrast, larger algae with well-defined morphological traits are easier to recognize using microscopic techniques [75]. In this study, microscopic observation identified a total of 79 phytoplankton species, while high-throughput sequencing covered a broader range of taxonomic groups, including Cryptophyta, Ochrophyta, and Rhodophyta, with approximately 601 OTUs.

Second, high-throughput sequencing is also subject to methodological biases. The rDNA gene copy numbers can vary significantly across different taxonomic groups, which may cause some groups to be overestimated or underestimated in relative abundance based on the sequencing data [76]. These variations can influence the estimation of relative abundances among different taxa.

Third, sample preservation and processing can also affect the results of the two methods. Microscopic observation depends on the integrity of cell morphology. If cells are damaged during sampling, fixation, or storage, certain taxonomic groups may not be accurately identified. By contrast, high-throughput sequencing is based on nucleic acid information and can detect DNA from dead or dormant cells, as well as extracellular DNA fragments. This may lead to overestimation of some taxa [34,75]. In addition, the differences in DNA extraction efficiencies among species may also introduce biases in the abundance estimates, resulting in discrepancies between the sequencing data and actual abundances in the environment [77,78,79,80] (Table 7).

Therefore, combining microscopic observation and high-throughput sequencing not only broadens our understanding of phytoplankton community structure but provides a basis for cross-validation, enhancing the scientific reliability of this study in analyzing phytoplankton diversity and ecological responses [33,81]. Although molecular biology techniques are rapidly advancing, microscopic identification methods remain an important tool for analyzing phytoplankton community structure.

## 5. Conclusions

This study used microscopic observation to evaluate the ecological effects of large-scale *Sargassum fusiforme* cultivation on coastal water quality and phytoplankton communities. The results demonstrated that, during the cultivation period, *S. fusiforme* significantly increased the pH and dissolved oxygen levels while reducing the concentrations of NO_3_–N, NO_2_–N, PO_4_–P, TP, and SiO_3_–Si, thereby alleviating coastal eutrophication and improving water quality. By contrast, during the non-cultivation period, the absence of *S. fusiforme* contributed to nutrient accumulation and weakened the water purification capacity.

Microscopic analysis further revealed that *S. fusiforme* cultivation enhanced the phytoplankton community diversity and stability, while significantly reducing the dominance of *Skeletonema costatum* in the cultivation area. These findings highlight the important role of *S. fusiforme* aquaculture in regulating coastal ecosystem functions.

It is recommended that future studies integrate microscopic observation with high-throughput sequencing to more comprehensively assess the effects of macroalgae cultivation on phytoplankton community structure, diversity, and ecological processes.

## Figures and Tables

**Figure 1 biology-14-00844-f001:**
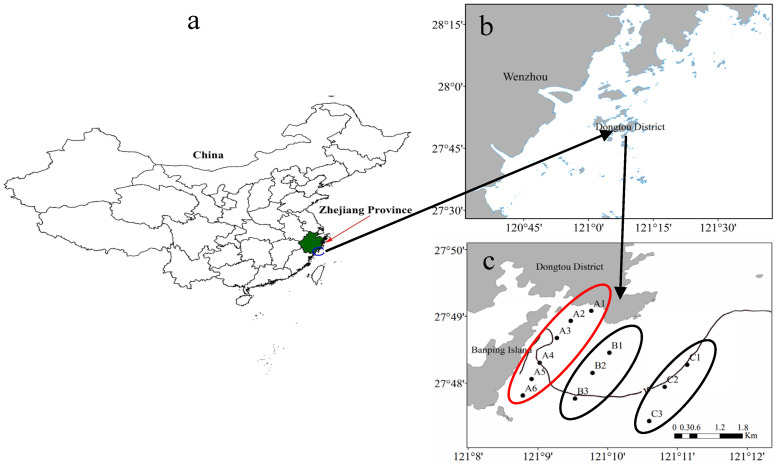
Sampling locations on Dongtou Island, China. (**a**) Location of Zhejiang Province in China. (**b**) Map showing Wenzhou and Dongtou District. (**c**) Distribution of sampling sites. A: cultivation area (stations A1–A6); B: adjacent area (stations B1–B3); C: control area (stations C1–C3).

**Figure 2 biology-14-00844-f002:**
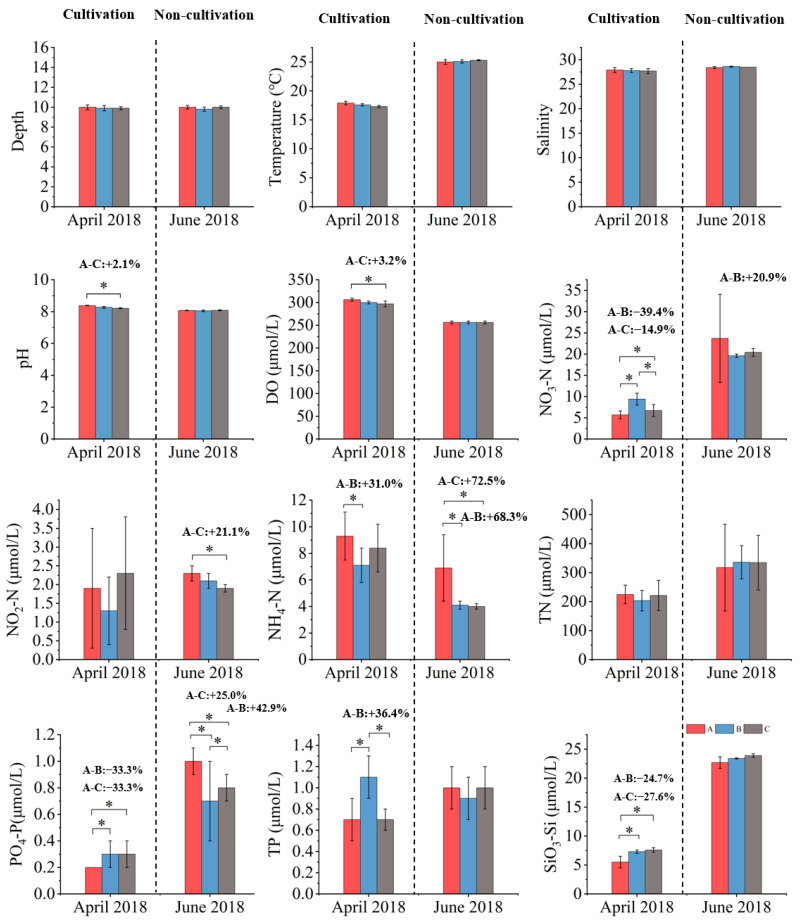
Temporal and spatial characteristics of environmental variables in 2018 (mean ± SD, *n* = 6). A: cultivation area, B: adjacent area, C: control area. (*) indicates significant differences (*p* < 0.05) between areas “B” or “C” and area “A”. Values on the figure indicate the relative percent decrease of the measured parameter in the cultivation area (A) compared to the adjacent area (B) and control area (C).

**Figure 3 biology-14-00844-f003:**
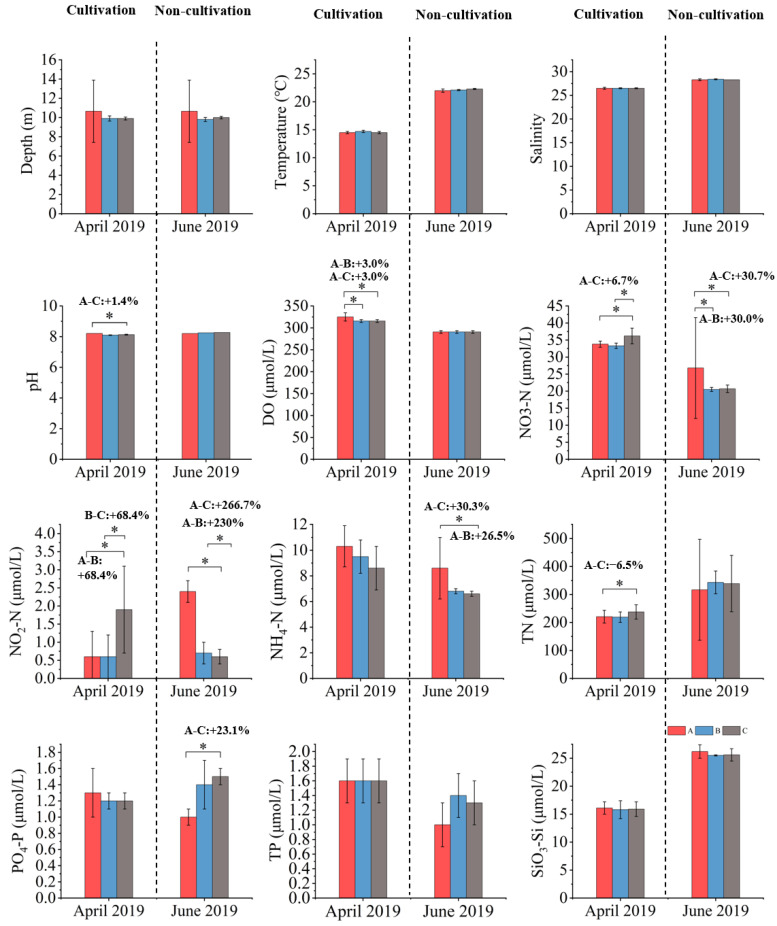
Temporal and spatial characteristics of environmental variables in 2019 (mean ± SD, *n* = 6). A: cultivation area, B: adjacent area, C: control area. (*) indicates significant differences (*p* < 0.05) between areas “B” or “C” and area “A”. Values on the figure indicate the relative percent decrease of the measured parameter in the cultivation area (A) compared to the adjacent area (B) and control area (C).

**Figure 4 biology-14-00844-f004:**
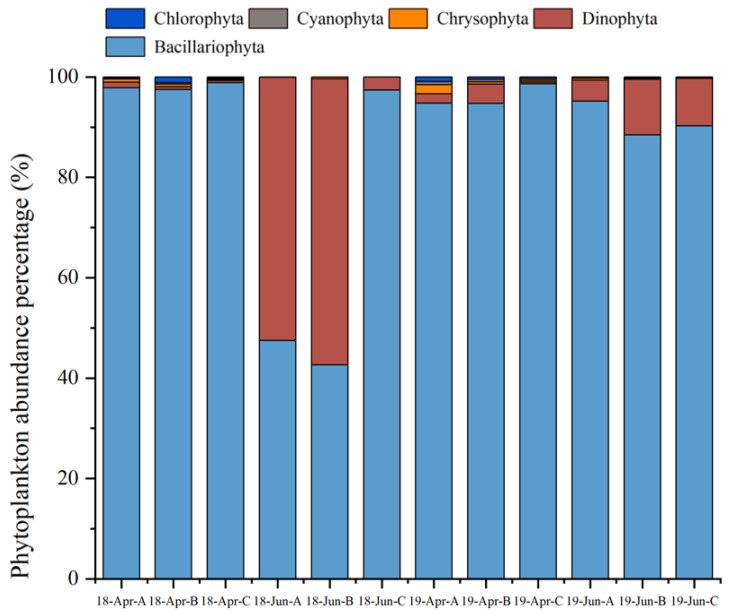
Phytoplankton abundance of each phylum at different sampling times (%). Note: Abbreviations follow the format “YY-MMM-X”, “YY” refers to the year (18 = 2018, 19 = 2019); “MMM” refers to the sampling month (Apr = April, Jun = June); “X” refers to the sampling area (A: cultivation area, B: adjacent area, C: control area).

**Figure 5 biology-14-00844-f005:**
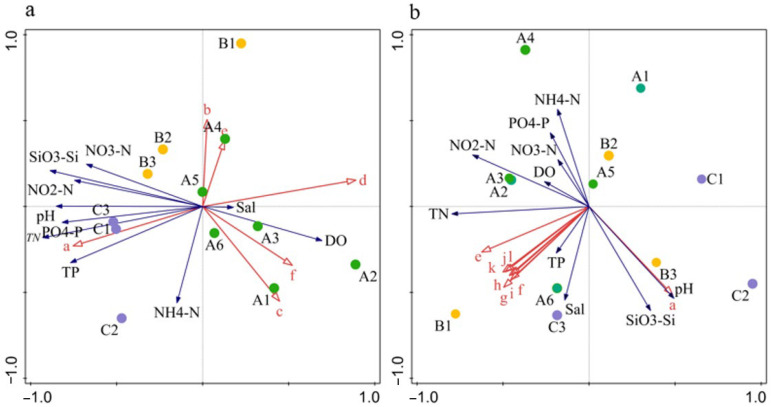
Biplot of phytoplankton dominant species density and environmental factors. a: *S. costatum*; b: *P. sulcata*; c: *C. jonesiana*; d: *P. blanda*; e: *P. pelagicum*; f: *S. acuminata*; g: *Alexandrium* sp.; h: *D. fibula*; i: *T. muelleri*; j: *T. furca*; k: *P. pentagonum*; l: *D. caudata.* Blue arrow: environmental factors; red arrow: dominant species of phytoplankton; solid circle: Station, A1−A6: cultivation area; B: B1–B3: adjacent area; C: C1−C3: control area. Panel (**a**): April 2018; panel (**b**): June 2018.

**Figure 6 biology-14-00844-f006:**
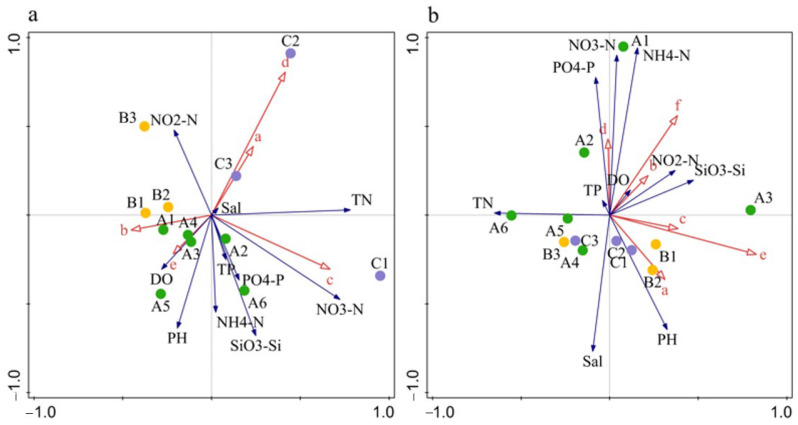
Biplot of phytoplankton dominant species density and environmental factors in 2019. a: *S. costatum*; b: *P. sulcata*; c: *C. jonesiana*; d: *P. blanda*; e: *P. pelagicum*; f: *S. acuminata*; g: *Alexandrium* sp.; h: *D. fibula*; i: *T. muelleri*; j: *T. furca*; k: *P. pentagonum*; l: *D. caudata*. Blue arrow: environmental factors; red arrow: dominant species of phytoplankton; solid circle: Station, A1–A6: cultivation area; B: B1–B3: adjacent area; C: C1–C3: control area. Panel (**a**): April 2019; panel (**b**): June 2019.

**Figure 7 biology-14-00844-f007:**
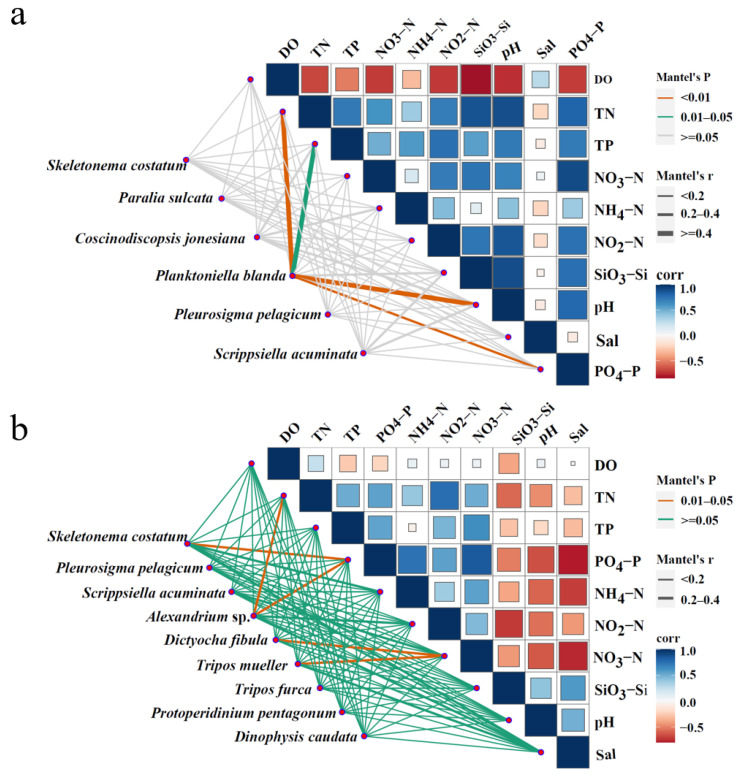
The association between environmental factors and dominant phytoplankton species based on a phylogenetic molecular ecological network analysis ((**a**): April 2018; (**b**): June 2018).

**Figure 8 biology-14-00844-f008:**
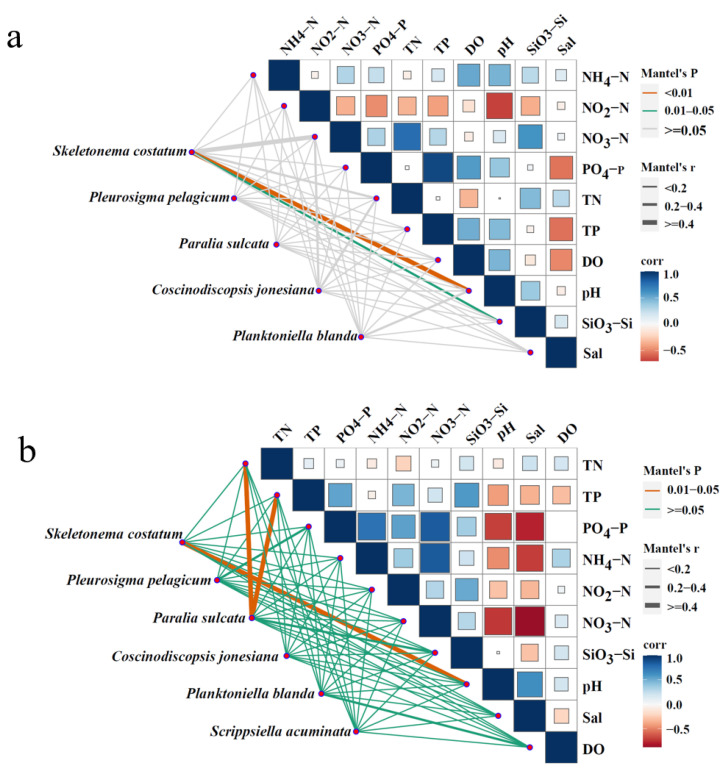
The association between environmental factors and dominant phytoplankton species based on a phylogenetic molecular ecological network analysis ((**a**): April 2019; (**b**): June 2019).

**Table 1 biology-14-00844-t001:** Phytoplankton abundance (cell/L) (mean ± SD, *n* = 6).

Area	April 2018	June 2018	April 2019	June 2019
A	1168 ± 564	15,982 ± 9531	632 ± 211	1522 ± 437
B	1167 ± 440	88,373 ± 10,855 *	660 ± 221	1300 ± 433
C	1172 ± 122	358,673 ± 136,877 *	640 ± 275	2090 ± 433 *

Note: *p* < 0.05. Significant differences were observed between groups B or C and group A, based on one-way ANOVA followed by Tukey’s HSD post hoc test. A: cultivation area, B: adjacent area, C: control area. “*” Indicates a significant difference compared with group A (*p* < 0.05).

**Table 2 biology-14-00844-t002:** Dominant phytoplankton species and their dominance values (Y) across sites in 2018 (mean ± SD, *n* = 6).

Phylum	Dominant Species	April			June				
		A	B	C	Mean	A	B	C	Mean
Bacillariophyta	*Skeletonema costatum* (Greville) Cleve 1873	0.17 *	0.38 *	0.74	0.37	0.45 *	0.41 *	0.97	0.83
	*Planktoniella blanda* (A.W.F.Schmidt) Syvertsen & Hasle 1993	0.26	0.15 *	0.01 *	0.14	-	-	-	-
	*Coscinodiscopsis jonesiana* (Greville) E.A.Sar & I.Sunesen 2008	0.21 *	0.08	0.06	0.14	-	-	-	-
	*Paralia sulcata* (Ehrenberg) Cleve, 1873	0.06	0.23 *	0.03	0.08	-	-	-	-
	*Pleurosigma pelagicum* Cleve 1894	0.04	0.10	-	0.03	0.01	-	-	-
Dinophyta	*Scrippsiella acuminata* (Ehrenberg) Kretschmann, Elbrächter, Zinssmeister, S.Soehner, Kirsch, Kusber & Gottschling 2015	0.01	0.01	0.02	0.01	0.47 *	0.44 *	0.02	0.13
	*Tripos muelleri* Bory 1826	-	-	-	-	0.01	0.02	-	-
	*Alexandrium* sp.	-	-	-	-	0.01	0.01	-	-
	*Tripos furca* (Ehrenberg) F.Gómez 2013	-	-	-	-	-	0.01	-	-
	*Protoperidinium pentagonum* (Gran) Balech 1974	-	-	-	-	-	0.01	-	-
	*Dinophysis caudata* Kent 1881	-	-	-	-	-	0.01	-	-
Chrysophyta	*Dictyocha fibula* Ehrenberg 1839	-	-	-	-	0.03	0.06	-	0.01

Note: “-”: *Y* < 0.01. *: *p* < 0.05 compared to the control group. A: cultivation area; B: adjacent area; C: control area.

**Table 3 biology-14-00844-t003:** Dominant phytoplankton species and their dominance values (Y) across sites in 2019 (mean ± SD, *n* = 6).

Phylum	Dominant Species	April			June				
		A	B	C	Mean	A	B	C	Mean
Bacillariophyta	*Skeletonema costatum* (Greville) Cleve 1873	0.06 *	0.13	0.19	0.12	0.36	0.38	0.46	0.38
	*Pleurosigma pelagicum* Cleve 1894	0.11	0.17	0.10	0.12	0.03	0.03	0.02	0.03
	*Paralia sulcata* (Ehrenberg) Cleve, 1873	0.18	0.18	0.17	0.18	0.02	0.02	0.01	0.02
	*Coscinodiscopsis jonesiana* (Greville) E.A.Sar & I.Sunesen 2008	0.16	0.20	0.23	0.19	0.23	0.23	0.26	0.24
	*Planktoniella blanda* (A.W.F.Schmidt) Syvertsen & Hasle 1993	0.07	0.20	0.25	0.20	0.24	0.13	0.20	0.20
	*Pinnularia* sp.	0.05	0.02	0.01	-	-	-	-	-
	*Thalassiosira eccentrica* (Ehrenberg) Cleve 1904	0.01	-	-	-	-	-	-	-
	*Chaetoceros lorenzianus* Grunow 1863	0.02	-	-	-	-	-	-	-
	*Synedra* sp.	-	0.02	-	-	-	-	-	-
	*Surirella* sp.	-	0.01	-	-	-	-	-	-
Dinophyta	*Scrippsiella acuminata* (Ehrenberg) Kretschmann, Elbrächter, Zinssmeister, S.Soehner, Kirsch, Kusber & Gottschling 2015	-	-	-	-	0.01	-	0.01	0.01

Note: “-”: *Y* ≤ 0.01. *: *p* < 0.05 compared to the control group. A: cultivation area; B: adjacent area; C: control area.

**Table 4 biology-14-00844-t004:** Phytoplankton diversity indices (mean ± SD, *n* = 6).

Diversity Indices	Area	April 2018	June 2018	April 2019	June 2019	*p*(S)
*H*′	A	1.566 ± 0.306	1.026 ± 0.187	1.826 ± 0.272	1.629 ± 0.231	*
	B	1.630 ± 0.309	1.235 ± 0.222	1.949 ± 0.163	1.501 ± 0.266	*
	C	0.826 ± 0.069 *	0.158 ± 0.052	1.501 ± 0.266 *	1.307 ± 0.125	*
*D*	A	1.025 ± 0.367	1.418 ± 0.187	1.221 ± 0.367	1.207 ± 0.387	*
	B	1.093 ± 0.403	1.322 ± 0.185	1.229 ± 0.346	1.070 ± 0.252	*
	C	0.643 ± 0.086 *	0.308 ± 0.066 *	1.031 ± 0.183 *	0.960 ± 0.143 *	*
*J*	A	0.764 ± 0.087	0.384 ± 0.073	0.884 ± 0.067	0.727 ± 0.058	*
	B	0.765 ± 0.068	0.456 ± 0.082	0.902 ± 0.046	0.699 ± 0.065	*
	C	0.477 ± 0.015 *	0.055 ± 0.019 *	0.657 ± 0.114 *	0.618 ± 0.026 *	*

Note: * *p* < 0.05. Significant differences were identified between B/C and A (one-way ANOVA and *t*-test). *p*(S) indicates significant differences between different times in the same sampling area (*p* < 0.05).

**Table 5 biology-14-00844-t005:** One-way ANOSIM results for phytoplankton community similarity analysis.

Group	ANOSIM	
	R	*p*
18-Apr-A vs. 18-Apr-B	0.272	0.119
18-Apr-A vs. 18-Apr-C	0.667	0.024
18-Jun-A vs. 18-Jun-B	0.105	0.631
18-Jun-A vs. 18-Jun-C	0.104	0.612
19-Apr-A vs. 19-Apr-B	0.160	0.214
19-Apr-A vs. 19-Apr-C	0.537	0.048
19-Jun-A vs. 19-Jun-B	0.105	0.274
19-Jun-A vs. 19-Jun-C	0.086	0.595

Note: Abbreviations follow the format “YY-MMM-X”, where “YY” indicates the year (18 = 2018, 19 = 2019), “MMM” indicates the sampling month (Apr = April, Jun = June), and “X” refers to the sampling area: A: cultivation area; B: adjacent area; C: control area.

**Table 6 biology-14-00844-t006:** Explanatory power (R^2^) and significance (*p*-value) of environmental variables based on the single-factor RDA and permutation tests (*n* = 999).

Time	Environmental Variable	R^2^	*p*	Significant
April 2018	SiO_3_;–Si	0.285	0.001	Significant
April 2018	TN	0.285	0.002	Significant
June 2018	NO_3_–N	0.194	0.065	Marginal
June 2019	SiO_3_–Si	0.165	0.056	Marginal
June 2019	NO_2_;–N	0.163	0.072	Marginal

Note: Variables with *p* < 0.05 are considered statistically significant; variables with 0.05 ≤ *p* < 0.10 are considered marginally significant.

**Table 7 biology-14-00844-t007:** Comparison of Microscopy and High-Throughput Sequencing for Phytoplankton Community Analysis.

Comparison Aspect	Microscopic Observation	High-Throughput Sequencing [33]
Identification Method	Identifies species based on morphological characteristics.	Uses 18S rRNA V4 region universal primers (18sV4F and 18sV4R) and Illumina MiSeq PE250/PE300 sequencing.
Accuracy of Phytoplankton Identification	Depends on morphological traits, making it difficult to distinguish morphologically similar species. Identifies Chlorophyta, Cyanophyta, Chrysophyta, Dinophyta, and Bacillariophyta.	Identifies up to genus or species level. Detects small, morphologically similar taxa including Bacillariophyta, Dinophyta, Chlorophyta, Cryptophyta, Chrysophyta, Ochrophyta, Rhodophyta, and Cyanophyta.
Number of Species and Dominant Species Identification	Identifies 75 species. Accurately quantifies larger-sized diatoms and dinoflagellate. Clearly reflects the influence of *S. costatum* dominance.	Detects approximately 601 OTUs. Identifies dominant species across a broad range of phytoplankton size classes, including microphytoplankton (>20 µm), nanophytoplankton (2–20 µm), and picophytoplankton (0.2–2 µm), and reveals taxonomic and abundance patterns
Abundance Metric	Uses cell counts (cells/L), which directly reflect actual abundance.	Uses OTU counts to reflect relative abundance of species.
Community Diversity and Stability	*H*′, *D*, and *J* indices in the cultivation and adjacent areas were higher, indicating that *S. fusiforme* cultivation enhances diversity and community stability.	α- and β-diversity indices indicate that *S. fusiforme* cultivation enhances community diversity and ecological stability.
Influence of Environmental Factors on Community Structure	N, P, Si, and DO are key environmental factors that influence community structure.	N, P, Si, and DO also influence community structure.
Time and Labor Cost	Requires taxonomic expertise. Time-consuming and labor-intensive.	Standardized process enables batch analysis and requires bioinformatics expertise.
Ecological Interpretation Capability	Reflects the spatial and temporal distributions of major species and helps interpret community dynamics.	Reveals taxon diversity and network structures, especially for difficult-to-identify taxa, providing insights into ecological processes.

## Data Availability

If required, the authors are willing to provide relevant data.

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
