# Peer review of "Ecological Effects of Sargassum fusiforme Cultivation on Coastal Phytoplankton Community Structure and Water Quality: A Study Based on Microscopic Analysis"

_biology, 2025, doi:10.3390/biology14070844_

Round 1
Reviewer 1 Report
Comments and Suggestions for Authors
A few general points:
1、When analyzing the differences in this paper, how was the control group determined? What is the purpose of dividing the sampling area into three different zones, A, B and C? Please clearly state it in the research methods.
2、Line168-182 “3.1 Water Quality”: What are the values of each water quality parameters before and after the cultivation of S. fusiforme, as well as in the planting area and non-planting area respectively? How are significant differences analyzed? Please reelaborate with data. The manifestations of the results of each water quality in Figure 2 do not highlight the significant differences between before and after cultivation, nor between the cultivation area and the non- cultivation area. Please redraw Figure 2.
3、Line 195-199: This paper takes April and June respectively as the comparison between cultivation and non-cultivation. Is this understanding correct? However, in the section of "Phytoplankton Abundance", there was no significant difference in phytoplankton abundance between the cultivation area and the non-cultivation area in April, but a significant difference was observed between the cultivation area and the non-cultivation area in June. Is it reasonable? Does it indicate that the cultivation of S. fusiforme has no effect on water quality?
4、Before providing reasonable explanations for the control groups before planting and in the non-planting areas, and before conducting quantitative discussions on the significant difference of "3.1 Water Quality", The conclusion drawn in section "4.1 S. fusiforme Cultivation Improves Water Quality" that the Cultivation of S. fusiforme can significantly improve water quality is not well-grounded.
5、This paper only conducted monitoring in April and June, and did not monitor different seasons. It is inappropriate to emphasize seasonal effects multiple times in "4.2 Impact of S. fusiforme Cultivation on Phytoplankton Community".
6、Since the research results in the paper are compared with those of high-throughput sequencing, it is necessary to provide a basic introduction to the high-throughput sequencing results and list the comparison of the research results of the two methods. Otherwise, others will not be clear about what kind of results they are.
More specific points:
- Lines 10–11: It is recommended to revise “microscopy-based” to “microscopy-assisted” or “microscopy-based quantitative enumeration” to more accurately reflect the meaning of “quantitative analysis by counting.”
- Lines 21–22: The phrase “thereby enhancing community diversity and ecosystem stability” is somewhat repetitive of earlier content and could be merged or rephrased for conciseness.
- Line 35: A spacing error is present; it should be “Sargassum fusiforme (Harvey) Setchell, 1931” with a space between the species name and the author citation.
- Lines 36–38: The description regarding changes in cultivation area lacks rigor. It is advised to provide the data source and clarify the assessment method used.
- Lines 60–61: The phrase “shaped by a range of...” is too vague. Consider specifying the key physical and chemical factors (e.g., nutrients, light availability).
- Lines 70–73: The sentence is overly long and could be split into two for improved readability and clarity.
- Explanations for ecological methods such as pMENs, RDA, and Monte Carlo tests are currently insufficient. Consider briefly elaborating on the rationale behind choosing these methods.
- Some reference numbers lack appropriate spacing or contain inconsistent punctuation (e.g., line 67). Please ensure formatting adheres to the journal’s reference style.
- Line 140: “ArcGIS 13.0” should be corrected to “ArcGIS 10.3” or a later officially released version for consistency and accuracy.
- Line 147: Remove the article “a” before “dominance values.”
- Line 162: “Environmental factor” should be corrected to the plural form “environmental factors.”
- Lines 164–165: The data processing description is clear, but it is important to specify whether normality (e.g., Shapiro–Wilk test) and homogeneity of variance (e.g., Levene’s test) were assessed to validate the ANOVA assumptions.
- If S. fusiforme is known to absorb nutrients, the observed high nutrient concentrations in some cultivation areas require explanation—whether they are transient or due to local inputs. It is suggested to further explore in the discussion whether such anomalies may result from rainfall runoff, sediment resuspension, or localized eutrophic discharge.
- Line 201: “p < 0.0.5” is an obvious typographical error and should be corrected.
- All instances of “p < 0.05” should be standardized to P < 0.05 with italicized P and consistent capitalization.
- Line 202: The description of significant differences is vague; please specify whether a post hoc test (e.g., Tukey HSD or LSD) was used or if differences were assessed via a t-test alone.
- Line 218: The notation “Y ≤ 0.01” should be standardized to “Y < 0.01” to align with common ecological reporting conventions.
- Line 236: The term “differed significantly” would benefit from the inclusion of the corresponding R-value and P-value to provide statistical support.
- Line 238: The description of results is redundant and could be streamlined for clarity and brevity.
- Species names such as S. costatum and S. acuminata should be written in full (genus and species) upon first mention, followed by abbreviated genus names in subsequent use, in accordance with ICBN conventions.
- Line 243: The punctuation “);” appears to be a typographical error and should be corrected for consistency.
- Line 278: Ensure a space follows every “Note:” for uniform formatting.
- Line 292: “Significan” is a typographical error and should be corrected to “significant.”
- Lines 257–259: The sample size (n) should be clearly indicated, and a discussion included on whether limited sample size may have contributed to nonsignificant results.
- Line 298: Use of the term “significant” should be more precise. If no multiple comparison correction was applied, consider stating “statistically significant based on unadjusted P-values.”
- Lines 328–337: The observed differences in dominant taxa between microscopy-based and high-throughput sequencing methods warrant further explanation. Consider discussing whether these discrepancies could stem from detection sensitivity related to cell size, methodological biases, or sample preservation effects.
Author Response
Comments 1: When analyzing the differences in this paper, how was the control group determined? What is the purpose of dividing the sampling area into three different zones, A, B and C? Please clearly state it in the research methods. |
Response 1: Thank you very much for your valuable comment.In this study, we determined the control group as the Control area C (stations C1–C3), which was located about 2000 meters offshore and considered to represent the natural background conditions without Cultivation disturbance. To assess the potential effects of S. fusiforme cultivation, we divided the sampling area into three zones according to their distance from the Cultivation area. The Cultivation area A (stations A1–A6) was located inside the farming area to evaluate the direct effects of cultivation on water quality. The Adjacent area B (stations B1–B3) was located 200–500 meters away from the farming area, where no farming was conducted, but this area may have been indirectly affected. This information is already included in the Methods section of the manuscript (Lines 129–135). Thank you again for your careful review and helpful feedback.
|
Comments 2: Line168-182 “3.1 Water Quality”: What are the values of each water quality parameters before and after the cultivation of S. fusiforme, as well as in the planting area and non-planting area respectively? How are significant differences analyzed? Please reelaborate with data. The manifestations of the results of each water quality in Figure 2 do not highlight the significant differences between before and after cultivation, nor between the cultivation area and the non- cultivation area. Please redraw Figure 2. |
Response 2: Thank you very much for your valuable comments. In response to your suggestions, we have revised Section 3.1 “Water Quality” (Lines 228–304) to clearly present the values of each water quality parameter before and after S. fusiforme cultivation, as well as in the cultivation, adjacent, and control areas. We applied one-way ANOVA and Tukey’s HSD post hoc tests to assess significant differences among sampling zones and periods, as described in the Methods section (Line186,188-191). In addition, we have redrawn Figure 2 (2018 data) and Figure 3 (2019 data) and added relative percentage changes and significance markers (*) to more clearly illustrate the differences among the sampling areas. We hope these revisions fully address your concerns, and we greatly appreciate your careful review and helpful suggestions. Comments 3: Line 195-199: This paper takes April and June respectively as the comparison between cultivation and non-cultivation. Is this understanding correct? However, in the section of "Phytoplankton Abundance", there was no significant difference in phytoplankton abundance between the cultivation area and the non-cultivation area in April, but a significant difference was observed between the cultivation area and the non-cultivation area in June. Is it reasonable? Does it indicate that the cultivation of S. fusiforme has no effect on water quality? Response 3: Thank you very much for your valuable comment. Your understanding is correct — in this study, April corresponds to the S. fusiforme cultivation period, while June corresponds to the post-harvest non-cultivation period. During the cultivation period in April, there were no significant differences in total phytoplankton abundance among the three areas. However, biodiversity indices (H′, D, and J) were significantly higher in the cultivation and adjacent areas than in the control area. The dominance of S. costatum was also significantly lower in the cultivation area. Specifically, its dominance decreased from 0.74 in the control area to 0.17 in the cultivation area in April 2018, and from 0.19 to 0.06 in April 2019. These results suggest that S. fusiforme cultivation may reduce the dominance of a single species and improve the evenness and diversity of the phytoplankton community, as described in the revised manuscript (Lines 558–566). In June, during the non-cultivation period, phytoplankton abundance increased significantly in the adjacent and control areas. The control area showed the highest abundance, and its community was strongly dominated by S. costatum, with a dominance reaching 0.97 in 2018, indicating a single-species dominated structure. In contrast, phytoplankton abundance in the cultivation area remained relatively low, and its community structure was more balanced. This difference may be related to the earlier regulatory effects of S. fusiforme cultivation on phytoplankton community structure or other environmental factors. Therefore, these findings do not imply that S. fusiforme cultivation has no impact on water quality. We appreciate your careful review and hope this explanation addresses your concern. Thank you again for your constructive feedback.
Comments 4: Before providing reasonable explanations for the control groups before planting and in the non-planting areas, and before conducting quantitative discussions on the significant difference of "3.1 Water Quality", The conclusion drawn in section "4.1 S. fusiforme Cultivation Improves Water Quality" that the Cultivation of S. fusiforme can significantly improve water quality is not well-grounded. Response 4: Thank you very much for your valuable comments.We have further improved the Methods and Results sections by providing detailed explanations for the selection of the control area and its representation of natural background conditions. In addition, in Section 3.1 “Water Quality,” we quantitatively compared water quality parameters across different areas and time periods using specific data and statistical analyses. These clearly show significant differences between the cultivation, adjacent, and control areas. Based on this data support, we believe that the conclusion in Section 4.1 “S. fusiforme Cultivation Improves Water Quality” is reasonable and well-founded. We will further strengthen the explanation and data support for this conclusion in the revised manuscript to ensure clear logic and sufficient evidence. Thank you again for your careful review and constructive suggestions.
Comments 5: This paper only conducted monitoring in April and June, and did not monitor different seasons. It is inappropriate to emphasize seasonal effects multiple times in "4.2 Impact of S. fusiforme Cultivation on Phytoplankton Community". Response 5: Thank you for your valuable comment. In response, we have revised the manuscript by replacing references to “seasonal effects” with “temporal variation” in Section 4.2. lines 531, 533, 562, 569, This change better reflects the limited monitoring period and avoids overemphasizing seasonal conclusions. We appreciate your careful review and helpful suggestions.
Comments 6: Since the research results in the paper are compared with those of high-throughput sequencing, it is necessary to provide a basic introduction to the high-throughput sequencing results and list the comparison of the research results of the two methods. Otherwise, others will not be clear about what kind of results they are. Response 6: Thank you very much for your valuable comments. We have carefully considered your suggestions and added a detailed comparison between microscopy and high-throughput sequencing methods in the revised manuscript. We included Table 5(line 677), which summarizes the differences and similarities between these two approaches in identification methods, accuracy, species detected, abundance metrics, diversity and stability, influence of environmental factors, time and labor costs, and ecological interpretation ability. This comparison helps readers better understand the strengths and limitations of each method and clarifies the significance of comparing our microscopic observations with high-throughput sequencing data. We sincerely appreciate your thorough review and valuable guidance and will continue to improve our manuscript.
Comments 7: Lines 10–11: It is recommended to revise “microscopy-based” to “microscopy-assisted” or “microscopy-based quantitative enumeration” to more accurately reflect the meaning of “quantitative analysis by counting.” Response 7: Thank you for your valuable suggestion. We have revised the phrase “microscopy-based” to “microscopy-based quantitative enumeration” in lines 25 to more accurately reflect the meaning of quantitative analysis by counting.
Comments 8: Lines 21–22: The phrase “thereby enhancing community diversity and ecosystem stability” is somewhat repetitive of earlier content and could be merged or rephrased for conciseness. Response 8: Thank you for your helpful comment. We agree that the phrase was repetitive. We have changed the sentence to make it shorter and clearer. Now it says: “It suppressed the dominance of Skeletonema costatum, promoting a more diverse and stable phytoplankton community.” This new version keeps the meaning but avoids repeating the same idea. The change is in the Abstract, Lines 33–34.
Comments 9: Line 35: A spacing error is present; it should be “Sargassum fusiforme (Harvey) Setchell, 1931” with a space between the species name and the author citation. Response 9: Thank you for pointing out this formatting oversight. We have corrected the spacing error in the revised manuscript. The species name now appears as “Sargassum fusiforme (Harvey) Setchell, 1931” with the appropriate space inserted between the binomial name and the author citation (Line 48).
Comments 10: Lines 36–38: The description regarding changes in cultivation area lacks rigor. It is advised to provide the data source and clarify the assessment method used. Response10: Thank you for your helpful comment. The data regarding the cultivation area of Sargassum fusiforme were obtained from the Dongtou Yearbook (2019), an official publication compiled by the local chronicles committee and research office of Dongtou District. This yearbook contains statistical information collected and verified by relevant government departments and is widely regarded as a reliable source of regional data. We also acknowledge a previous error in the manuscript regarding the reported cultivation area. In an earlier version, the area was incorrectly stated as over 13,000 hectares due to a unit conversion mistake. The original figure was 13,000 mu, which should have been correctly converted to approximately 867 hectares (1 hectare = 15 mu). This error has now been corrected in the revised manuscript (Line 52).
Comments 11: Lines 60–61: The phrase “shaped by a range of...” is too vague. Consider specifying the key physical and chemical factors (e.g., nutrients, light availability). Response 11: We appreciate the reviewer’s detailed suggestion. In response, we have revised the sentence to specify the key physical and chemical factors in order to enhance clarity and scientific precision. The revised sentence now reads: “The composition and diversity of phytoplankton communities are strongly influenced by environmental changes and are shaped by a combination of physical (e.g., temperature, light intensity, salinity) and chemical (e.g., nutrient concentrations such as nitrogen and phosphorus) factors.” This change has been incorporated into Lines74–76 of the revised manuscript.
Comments 12: Lines 70–73: The sentence is overly long and could be split into two for improved readability and clarity. Response 12: Thank you for your helpful suggestion. In response, we have revised the sentence to improve readability and clarity by splitting it into two shorter sentences. The revised text now reads:
Comments 13: Explanations for ecological methods such as pMENs, RDA, and Monte Carlo tests are currently insufficient. Consider briefly elaborating on the rationale behind choosing these methods. Response 13: Thank you for pointing this out. In response to your suggestion, we have added brief explanations of the rationale for selecting each ecological method in the revised manuscript. Specifically: RDA was chosen after detrended correspondence analysis (DCA) indicated that the gradient lengths were all less than 3, suggesting that a linear ordination model was appropriate. RDA allowed us to visualize and quantify the relationships between phytoplankton communities and environmental variables. The Monte Carlo permutation test (n = 999) was used to assess the statistical significance of the RDA model. As a non-parametric test, it is particularly suitable for small ecological datasets and helps confirm that the observed relationships are not due to random variation. Phylogenetic Molecular Ecological Network analysis (pMENs) was employed to further explore co-occurrence patterns and potential ecological interactions between dominant phytoplankton groups and key environmental variables. This method also helped identify network properties (e.g., connectivity, modularity) and key environmental drivers. These additions have been incorporated into the revised manuscript (Lines 190–211) to enhance clarity and methodological transparency.
Comments 14: Some reference numbers lack appropriate spacing or contain inconsistent punctuation (e.g., line 67). Please ensure formatting adheres to the journal’s reference style. Response 14: Thank you for your careful observation. We have carefully reviewed all reference numbers and citations throughout the manuscript. Formatting inconsistencies, including missing spaces and incorrect punctuation (such as those noted in line 67), have been corrected to ensure full compliance with the journal’s reference style guidelines. These changes have been incorporated in the revised manuscript. Comments 15: Line 140: “ArcGIS 13.0” should be corrected to “ArcGIS 10.3” or a later officially released version for consistency and accuracy. Response 15: Thank you for pointing out this error. We have corrected “ArcGIS 13.0” to the officially released version “ArcGIS 10.3” in the revised manuscript to ensure consistency and accuracy (Line171).
Comments 16: Line 147: Remove the article “a” before “dominance values.” Response 16: Thank you for your careful review. We have removed the article “a” before “dominance values” in the revised manuscript, as suggested (Line178). Comments 17: Line 162: “Environmental factor” should be corrected to the plural form “environmental factors.” Response 17: Thank you for pointing this out. We have corrected “environmental factor” to the plural form “environmental factors” in line 212, 432,444.
Comments 18: Lines 164–165: The data processing description is clear, but it is important to specify whether normality (e.g., Shapiro–Wilk test) and homogeneity of variance (e.g., Levene’s test) were assessed to validate the ANOVA assumptions. Response 18: Thank you for your valuable comment. In response, we have added clarification to indicate that both normality and homogeneity of variance were tested prior to ANOVA. Specifically, the following sentence has been added to the revised manuscript: “Prior to ANOVA, the Shapiro–Wilk test was used to assess data normality, and Levene’s test was performed to evaluate the homogeneity of variances. Only when both assumptions were satisfied (p > 0.05), Tukey’s HSD post hoc test was conducted.” This revision has been incorporated into Lines 185-188. Comments 19: If S. fusiforme is known to absorb nutrients, the observed high nutrient concentrations in some cultivation areas require explanation—whether they are transient or due to local inputs. It is suggested to further explore in the discussion whether such anomalies may result from rainfall runoff, sediment resuspension, or localized eutrophic discharge. Response 19: Thank you for your insightful comment. In response, we have expanded the discussion to address the observed nutrient anomalies in certain cultivation areas. We acknowledge that although S. fusiforme has nutrient absorption capacity, short-term increases in nutrient concentrations may occur due to external factors such as rainfall runoff, sediment resuspension, and nearshore discharges. For instance, we noted slightly elevated NH₄-N and TN levels in the cultivation area during the farming period (April 2018), which may reflect such localized influences. We also cited relevant literature (e.g., Howarth, 2008) to support this explanation. These additions provide a more comprehensive understanding of the temporal and spatial variability in nutrient concentrations and appear in the revised manuscript (Lines 500–509).
Comments 20: Line 201: “P < 0.0.5” is an obvious typographical error and should be corrected. Response 20: Thank you for pointing out this typographical error. We have corrected “P < 0.0.5to “P < 0.05” in the revised manuscript (Line 339). Comments 21: All instances of “P < 0.05” should be standardized to P < 0.05 with italicized P and consistent capitalization. Response 21: Thank you for your careful observation. We have reviewed the entire manuscript and standardized all instances of “P < 0.05” to “P < 0.05,” with italicized and capitalized P, in accordance with the journal’s formatting guidelines. Comments 22: Line 202: The description of significant differences is vague; please specify whether a post hoc test (e.g., Tukey HSD or LSD) was used or if differences were assessed via a t-test alone. Response 22: Thank you for your valuable comment. We have clarified the statistical procedure in both the Methods section and the table note. As described in the Methods section, one-way ANOVA was used to evaluate differences in phytoplankton abundance among sampling areas. Prior to ANOVA, the Shapiro–Wilk and Levene’s tests were applied to assess normality and homogeneity of variances. When both assumptions were met (P > 0.05), Tukey’s HSD post hoc test was conducted to identify significant pairwise differences. This information is now clearly reflected in the revised note of Table 1 (Line339-341) and the relevant text in the Methods section (Lines 185–188).
Comments 23: Line 218: The notation “Y ≤ 0.01” should be standardized to “Y < 0.01” to align with common ecological reporting conventions. Response 23: Thank you for your observation. We have revised the notation from “Y ≤ 0.01” to “Y < 0.01” in the revised manuscript (Line357) to conform with standard ecological reporting practices.
Comments 24: Line 236: The term “differed significantly” would benefit from the inclusion of the corresponding R-value and P-value to provide statistical support. Response 24: We appreciate the reviewer’s suggestion. In the revised manuscript, we have added the relevant R and P values to clarify and support the description of community differences. Specifically, the ANOSIM result in April 2018 between the cultivation and control areas was R = 0.667, P = 0.024. These details have been incorporated into the main text (Lines 390–391) and are also reported in Table 5.
Comments 25: Line 238: The description of results is redundant and could be streamlined for clarity and brevity. Response 25: Thank you for the suggestion. We have revised the sentence in the Results section to improve clarity and remove redundancy. The updated version now reads: Comments 26: Species names such as S. costatum and S. acuminata should be written in full (genus and species) upon first mention, followed by abbreviated genus names in subsequent use, in accordance with ICBN conventions. Response 26: Thank you for your reminder. We confirm that the full species names were provided upon their first mention in the main text, in accordance with ICBN conventions. Specifically, Skeletonema costatum first appears in Line 343, Paralia sulcata first appears in Line 459, Coscinodiscopsis jonesiana first appears in Line 637, Pleurosigma pelagicum first appears in Line 638, Scrippsiella acuminata first appears in Line 348, Tripos muelleri first appears in Line 456, Protoperidinium pentagonum first appears in Line 402, Dinophysis caudata first appears in Line 595, Dictyocha fibula first appears in Line 402. In subsequent mentions, abbreviated genus names are used appropriately. Comments 27: Line 243: The punctuation “);” appears to be a typographical error and should be corrected for consistency. Response27:Thank you for pointing this out. We have corrected the punctuation error in Line 397 by removing the unnecessary semicolon to ensure consistency with the rest of the manuscript.
Comments 28: Line 278: Ensure a space follows every “Note:” for uniform formatting. Response 28: Thank you for your suggestion. We have reviewed all instances of “Note:” in the manuscript and corrected the formatting by adding a space after “Note:” where necessary, including in Line 339, 363, 385, 395 to ensure consistency throughout the text. Comments 29: Line 292: “Significan” is a typographical error and should be corrected to “significant.” Response 29: Thank you for pointing this out. We have corrected the typographical error in Line 448 by changing “Significan” to “significant” in the revised manuscript. Comments 30: Lines 257–259: The sample size (n) should be clearly indicated, and a discussion included on whether limited sample size may have contributed to nonsignificant results. Response 30: Thank you for pointing this out. We have now clearly indicated the sample size in the revised manuscript as “n = 6 per group”, and we have added a discussion on the potential influence of limited sample size on the statistical power of the RDA models. Specifically, we acknowledge that the relatively small sample size may have reduced the ability to detect significant multivariate relationships. The revised sentence now reads: “This may be partly attributed to the relatively limited sample size (n = 6 per group), which could reduce the statistical power to detect significant multivariate relationships.” This revision appears in Lines 410-412 of the revised manuscript. Comments 31: Line 298: Use of the term “significant” should be more precise. If no multiple comparison correction was applied, consider stating “statistically significant based on unadjusted P-values.” Response 31: Thank you very much for your insightful comment. We appreciate your attention to the precise use of the term “significant.” As you rightly pointed out, in our study, statistical significance was assessed without applying multiple comparison correction. In response, we have revised the relevant sentence to clarify that the significance is based on unadjusted P-values. Specifically, the sentence now reads (Lines 451–452): “… were analyzed using pMENs based on Mantel’s tests. The significance of correlations was determined from unadjusted P-values (P < 0.05).” Comments 32: Lines 328–337: The observed differences in dominant taxa between microscopy-based and high-throughput sequencing methods warrant further explanation. Consider discussing whether these discrepancies could stem from detection sensitivity related to cell size, methodological biases, or sample preservation effects. Response 32: Thank you for pointing this out. we have added a detailed explanation regarding the observed differences in dominant taxa between microscopy-based and high-throughput sequencing methods (Lines 664–676, 680-688): First, detection sensitivity is closely related to cell size. High-throughput sequencing can efficiently detect picoplankton and nanoplankton. These small phytoplankton often cannot be reliably identified under traditional light microscopy due to their tiny size and lack of distinct morphological features. In contrast, larger algae with well-defined morphological traits are easier to recognize using microscopic techniques [75]. In this study, microscopic observation identified a total of 79 phytoplankton species, while high-throughput sequencing covered a broader range of taxonomic groups, including Cryptophyta, Ochrophyta, and Rhodophyta, with approximately 601 OTUs. Second, high-throughput sequencing is also subject to methodological biases. rDNA gene copy numbers can vary significantly across different taxonomic groups, which may cause some groups to be overestimated or underestimated in relative abundance based on sequencing data [76]. These variations can influence the estimation of relative abundances among different taxa. Third, sample preservation and processing can also affect the results of the two methods. Microscopic observation depends on the integrity of cell morphology. If cells are damaged during sampling, fixation, or storage, certain taxonomic groups may not be accurately identified. In contrast, high-throughput sequencing is based on nucleic acid information and can detect DNA from dead or dormant cells, as well as extracellular DNA fragments. This may lead to overestimation of some taxa [34, 75). In addition, differences in DNA extraction efficiencies among species may also introduce biases in abundance estimates, resulting in discrepancies between sequencing data and actual abundances in the environment [77-80] (Table 5).
|
4. Response to Comments on the Quality of English Language |
Point 1: The English could be improved to more clearly express the research Response: We sincerely thank the reviewer for the valuable comment on the English language. In response, we carefully revised the whole manuscript to improve clarity, grammar, and readability. We checked for spelling mistakes, awkward expressions, inconsistencies, and long or complex sentences. We tried our best to make the language more accurate and easier to understand. |
|
5. Additional clarifications |
Thank you for your suggestion. We have added the necessary clarifications in the revised manuscript to improve understanding. The relevant section has been modified to provide more detailed explanation
|

Reviewer 2 Report
Comments and Suggestions for Authors
The manuscript entitled “Ecological Effects of Sargassum fusiforme Cultivation on Coastal Phytoplankton Community Structure and Water Quality: A Study Based on Microscopic Analysis” is devoted to microscopy-based quantitative analysis investigation of the long-term dynamic effects of two consecutive years of Sargassum fusiforme cultivation on phytoplankton diversity and water quality. The authors use several approaches in their work, including traditional microscopy, high-throughput sequencing combined with redundancy analysis (RDA) 68 and phylogenetic molecular ecological networks (pMENs). This study provides valuable insights into the potential ecological impacts of S. fusiforme cultivation on marine ecosystems and offers important guidance for the ecological management of sustainable aquaculture in the future. To my mind this manuscript is corresponding to the aims and scopes of the Biology journal. I am ready to recommend it for publication after corrections, due to the comments below.
- In the abstract, it is worth explaining what exactly is meant by water quality
- 53-54 general information that I advise to shorten
- It is worthwhile to describe the purpose of the study more clearly
- It is worthwhile to pay more detailed attention to ecological management in the introduction
- 96 What hydrological changes are we talking about?
- It is necessary to separate analytical methods separately, describing in detail the features of the analysis and the manufacturer of the device. I did not notice any information about sequencing and primers
- Fig. 2 shows data on salinity. It is necessary to clarify the reasons for its change
- The authors are studying the SiO3-Si factor, it is worth explaining why it is important.
- It would be interesting to use a factor that does not change over time in order to compare it with the nature of changes in biologically mediated factors
- Are phosphorus sources added during cultivation? Then the fact that cultivation has a positive effect on its reduction is somewhat obvious.
- It would be interesting to hear the authors' comments on whether they have found any species dangerous to human health. Cyanobacteria?
Author Response
1. Comments 1: In the abstract, it is worth explaining what exactly is meant by water quality |
Response 1: Thank you for your suggestion. We have added a clear explanation in the abstract about what we mean by “water quality.” We listed the main parameters we measured in this study, such as pH, dissolved oxygen (DO), nitrate nitrogen (NO3-N), nitrite nitrogen (NO2-N), phosphate phosphorus (PO4-P), total phosphorus (TP), and silicate silicon (SiO3-Si), in lines 28-30. This helps readers better understand which aspects of water quality we assessed. We appreciate your careful review and valuable comments.
|
2. Comments 2: 53-54 general information that I advise to shorten Response 2: Thanks for the suggestion. We’ve shortened the text in Lines 67 and now simply state that “Phytoplankton play an essential role in marine primary production” to keep it clear and concise.
|
3. Comments 3: It is worthwhile to describe the purpose of the study more clearly Thank you very much for your valuable suggestion. Response 3: In response to your comment, we have clarified the purpose of the study both in the Abstract (Lines 25–37) and at the end of the Introduction (Lines 92–97). These revisions emphasize that our study aims to evaluate the ecological effects of large-scale S. fusiforme cultivation on coastal phytoplankton communities and water quality, using microscopy-based enumeration for more accurate quantitative assessment. We also specify the use of data collected in April and June of 2018 and 2019 to capture spatial and temporal variations. We hope these additions help present the research purpose more clearly. We sincerely appreciate your thoughtful feedback, which has helped improve the quality of our manuscript.
Comments 4: It is worthwhile to pay more detailed attention to ecological management in the introduction Response 4: Thank you for your suggestion. To address this point, we have added a short paragraph in the Introduction (Lines 79–86) to more clearly emphasize the importance of ecological management in seaweed cultivation. This new content briefly discusses the potential ecological risks of large-scale farming and highlights why understanding the response of phytoplankton communities is important for guiding sustainable practices. We hope this addition helps clarify the ecological context of our work. The added paragraph reads: “Therefore, phytoplankton are not only key primary producers but also important biological indicators for assessing changes in water environments and ecosystem health. With the rapid expansion of macroalgae farming, there is a growing need to manage its ecological impacts more effectively. Ecological management in seaweed cultivation involves not only maintaining stable yields, but also ensuring that farming practices do not disrupt local biodiversity, nutrient balance, or water quality. In particular, understanding how S. fusiforme cultivation influences phytoplankton communities can help identify early signs of ecological imbalance and guide sustainable farming decisions.”
Comments 5: 96 What hydrological changes are we talking about? Response 5: Thank you very much for your valuable comment. In response to your question about the phrase "hydrological changes," we have revised the sentence in Lines 119–121 of the manuscript to clarify what this term includes. The revised sentence now reads: “This approach also allowed us to assess how seasonal climate and hydrological changes, such as rainfall, runoff, salinity fluctuations, etc., may influence phytoplankton dynamics.” By using “such as… etc.,” we aim to indicate that these changes include, but are not limited to, the listed factors. This broader phrasing helps to capture the range of possible environmental influences on phytoplankton community structure. We appreciate your careful review and hope this clarification addresses your concern.
4. Comments 6: It is necessary to separate analytical methods separately, describing in detail the features of the analysis and the manufacturer of the device. I did not notice any information about sequencing and primers Response 6: Thank you for your helpful comment. In response, we have added a new subsection “2.4. Reference Information on High-Throughput Sequencing” lines 214-220 to provide clarification. Since the high-throughput sequencing data used in this study were obtained from our previously published work [33], we have now clearly stated the sequencing platform (Illumina MiSeq PE250/PE300), primer set (18sV4F and 18sV4R), and indicated that full technical details are available in that publication. In the current manuscript, these data are used only for comparison purposes to support the microscopy-based analysis. We hope this addition resolves your concern. Thank you again for your valuable suggestion.
5. Comments 7: Fig. 2 shows data on salinity. It is necessary to clarify the reasons for its change Response 7: Thank you for your suggestion. We have now explained the seasonal variation in salinity in the revised Discussion. Specifically, we noted that salinity rose from April to June in both years, which is likely related to lower rainfall and higher evaporation during the early summer months. This change typically reflects natural seasonal patterns in coastal areas. The relevant explanation has been added to the Discussion section, Lines 513–515.
Comments 8: The authors are studying the SiO3-Si factor, it is worth explaining why it is important. SiO3-Si is a key nutrient that supports the growth of siliceous phytoplankton, especially diatoms, which are major contributors to coastal primary production. Changes in SiO3-Si availability can influence phytoplankton community composition and succession patterns. In our study, RDA results from April 2018 showed that SiO3-Si was significantly correlated with phytoplankton community structure (R2 = 0.285, P = 0.001), highlighting its ecological role. Although S. fusiforme does not directly utilize silicate, the observed changes in SiO3-Si concentrations may reflect indirect effects of cultivation on phytoplankton dynamics and nutrient cycling. In response to the reviewer’s comment, we have added a clear explanation of the importance of SiO3-Si in the Discussion section (Lines 609–617).
6. Comments 9: It would be interesting to use a factor that does not change over time in order to compare it with the nature of changes in biologically mediated factors Response 9: Thank you for your suggestion. In natural marine environments, most environmental factors—such as temperature, salinity, nutrient concentrations, and dissolved oxygen—vary over time due to seasonal changes, hydrodynamics, and biological activity. Therefore, it is difficult to identify a completely time-invariant factor. However, we agree that using relatively stable references, such as water depth or fixed geographic location, can help in comparing with biologically mediated changes. In our study, the sampling was conducted at fixed geographic locations, which served as a consistent reference throughout the investigation.
7. Comments 10: Are phosphorus sources added during cultivation? Then the fact that cultivation has a positive effect on its reduction is somewhat obvious. Response10: Thank you for your comment. No phosphorus sources were added during the cultivation period. Therefore, the observed reduction in phosphate concentration is likely due to the uptake by S. fusiforme and natural environmental processes, rather than external nutrient input.
8. Comments 11: It would be interesting to hear the authors' comments on whether they have found any species dangerous to human health. Cyanobacteria? Response 11: Thank you for your valuable question. Based on our microscopic observations, we did not detect any cyanobacteria in the phytoplankton samples during the study period. However, we did observe a few dinoflagellate species with known toxic potential, including Alexandrium sp. and Dinophysis caudata, which are well-documented producers of PSP and DSP toxins, respectively [65-66]. These species were recorded at low abundance, and no harmful algal bloom (HAB) events were observed. Therefore, we believe that S. fusiforme cultivation did not facilitate the proliferation of toxin-producing species and may even help mitigate HAB risks by maintaining higher phytoplankton diversity and ecosystem stability. A brief description of these findings has been added to the Discussion section (Lines 592-600) of the revised manuscript.
|
4. Response to Comments on the Quality of English Language |
Point 1: No specific comments were raised regarding the English language. We have nonetheless carefully proofread the manuscript to ensure clarity, correctness, and consistency throughout the text. |
|
5. Additional clarifications |
Thank you for your suggestion. We have added the necessary clarifications in the revised manuscript to improve understanding. The relevant section has been modified to provide more detailed explanation
|

Reviewer 3 Report
Comments and Suggestions for Authors
The article by Yurong Zhang and co-authors is of scientific interest, presenting sound material collected in the field over two years. The main idea of the article is also interesting – the influence of seaweed cultivation systems on phytoplankton. The authors sought to identify the primary mechanisms underlying the interaction between two systems that represent the primary productive part of the ecosystem. In general, the article can be published after several shortcomings have been eliminated. The main disadvantage of this work is the lack of information about the dynamics of waters, in particular about the direction and speed of currents. This information could clarify some of the effects of the interaction between the two systems. However, such information is missing; otherwise, the authors would have provided it.
I recommend that the authors refine the Abstract, as it currently leaves the impression of being incomplete. Even the Conclusions are more concise and coherent. There are some comments on the text.
Line 47 – need a reference;
Line 114 – "gallon bucket", it needs CI;
Line150 – number or abundance?;
Line 170 – here, you need a ref to Figure 2;
Lines 293-302. There is no explanation for why the abundance of Sk costatum correlates with silicon when its concentration is high; moreover, seaweed does not consume silicon.
Figure 2. It is advisable to use µM here and in the text. The concentrations of nitrogen and silicon are very high. At such concentrations, there should be a bloom of phytoplankton with high biomass. We need to check the measurements and calculations.
Figures 6 and 7 are difficult to read because of the small print.
Reference
There is a lot of carelessness here, for example, 496, 582, 591 (another author of the article is not specified), etc.
[22] – does not match the text, it is advisable to replace;
[39-41] – unsuccessful references, as they do not relate to the open sea area.
Author Response
Comments 1: Line 47 – need a reference; Response 1: Thank you for pointing this out. We have added appropriate citations at Line 60 to support the statement, specifically referencing [17] and [18]. |
|
Comments 2: Line 114 – "gallon bucket", it needs CI; Response 2: Thank you for the suggestion. We have revised “gallon bucket” to “2.5 L bucket” in line 144 to align with international unit standards.
|
Comments 3: Line150 – number or abundance? Response 3: Thank you for the helpful comment. We have replaced “number” with “abundance” in Line 180 to better reflect the intended ecological context.
Comments 4: Line 170 – here, you need a ref to Figure 2; Response 4: Thank you for your helpful reminder. We have revised the Results section “3.1 Water Quality” (Lines 228–304) to present detailed values for each parameter across different periods and zones. In addition, clear references to Figures 2 and 3 have been added at the appropriate positions in the revised text. We have also redrawn Figure 2 (2018 data) and Figure 3 (2019 data), incorporating relative percentage changes and significance markers (*) to more clearly illustrate the differences among the sampling areas.
Comments 5: There is no explanation for why the abundance of S. costatum correlates with silicon when its concentration is high; moreover, seaweed does not consume silicon. Response 5: Thank you for your insightful comment. We agree that Sargassum fusiforme does not directly utilize silicate. However, Skeletonema costatum is a diatom that requires silicon for frustule formation, and its abundance is often closely linked to SiO₃-Si availability. In our study, the observed correlation between S. costatum abundance and SiO₃-Si concentration reflects this ecological requirement. As described in the revised manuscript (Lines 609–617), SiO₃-Si serves as a limiting nutrient for siliceous phytoplankton. Therefore, the high silicon levels may support diatom growth, including S. costatum, rather than being related to seaweed uptake. This explanation has been added to the Discussion section to clarify the ecological relationship.
Comments 6: Figure 2. It is advisable to use µM here and in the text. The concentrations of nitrogen and silicon are very high. At such concentrations, there should be a bloom of phytoplankton with high biomass. We need to check the measurements and calculations. Response 6: We thank you for this valuable comment. To ensure clarity and consistency, we have revised the concentration units in Figures 2 and 3 and throughout the text to µM (µmol/L). We have carefully re-checked all measurements and calculations related to nitrogen and silicon concentrations and confirmed their accuracy. Although the observed levels were relatively high in some cases, they still fall within the background concentration ranges reported for the nearshore East China Sea. According to Ye et al. (2020), typical spring and summer values of DIN, DIP, and DSi in this region range from 5.3–11.9 µM, 0.31–0.53 µM, and 11.1–23.2 µM, respectively. The values in our study are within these ranges, indicating that the nutrient concentrations reflect natural regional conditions. Moreover, high nutrient levels did not always correspond with high phytoplankton biomass in our data. This suggests that nutrient enrichment alone does not necessarily lead to bloom formation. Other environmental factors, such as water temperature, also play important roles. For instance, the highest phytoplankton abundance was observed in June 2018 when the water temperature reached ~25°C, while no significant increase was seen in other periods with lower temperatures, despite similarly high nutrient concentrations. This indicates that suitable thermal conditions, in combination with nutrient availability, are necessary to trigger bloom events in this coastal system. We have incorporated this discussion into Section 4.1 (lines 481-497 of the revised manuscript) to better clarify the relationship between nutrient levels and bloom occurrence.
Comments 7: Figures 6 and 7 are difficult to read because of the small print. Response 7: Thank you for your helpful suggestion. We have adjusted Figures 6 and 7 by increasing the font size of the axis labels, legends, and other text elements to enhance readability. The revised figures are now easier to read and have been updated in the manuscript accordingly.
Comments 8: Reference There is a lot of carelessness here, for example, 496, 582, 591 (another author of the article is not specified), etc. Response 8: Thank you for pointing out this issue. We have carefully reviewed and corrected all reference formatting problems in the revised manuscript. Missing author information and other inconsistencies have been fixed to ensure full compliance with the journal's reference style requirements.
Comments 9: [22] – does not match the text, it is advisable to replace; Response 9: Thank you for pointing this out. We’ve reviewed the citation and agree that the previous reference was not a good fit for the content. It has been replaced with a more appropriate and current source that better supports the statement.
Comments 10: [39-41] – unsuccessful references, as they do not relate to the open sea area. Response10: Thank you for pointing this out. After reviewing the literature, the authors found that there is no fixed threshold for phosphate concentration required for phytoplankton growth in the ocean. The critical level varies depending on several factors. These include the type of phytoplankton, such as diatoms, dinoflagellates, or cyanobacteria. It also depends on the water type, for example, coastal or open sea. Environmental conditions like temperature, light, and the nitrogen to phosphorus ratio also play a role. In addition, background nutrient status, whether eutrophic or oligotrophic, affects phosphate demand. These factors together shape the response of phytoplankton to phosphorus. As a result, the threshold differs across ecosystems. Therefore, the conclusion that phosphorus limitation strongly suppressed phytoplankton growth is not well supported. We have removed the related discussion in lines 100 to 105 from the revised manuscript. Consequently, references [39-41] have also been deleted.
|
4. Response to Comments on the Quality of English Language |
Point 1: No specific comments were raised regarding the English language. We have nonetheless carefully proofread the manuscript to ensure clarity, correctness, and consistency throughout the text. |
|
5. Additional clarifications |
Thank you for your suggestion. We have added the necessary clarifications in the revised manuscript to improve understanding. The relevant section has been modified to provide more detailed explanation
|

Round 2
Reviewer 2 Report
Comments and Suggestions for Authors
To my opinion, the authors have significantly revised the manuscript and taken into account all my comments. I am ready to recommend the manuscript for publication in this form.